# Comprehensive Reassessment of Large-Scale Evaluation Outcomes in LLMs: A Multifaceted Statistical Approach

## Abstract

Amidst the rapid evolution of LLMs, the significance of evaluation in comprehending and propelling these models forward is increasingly paramount. Evaluations have revealed that factors such as scaling, training types, architectures and other factors profoundly impact the performance of LLMs. However, the extent and nature of these impacts continue to be subjects of debate because most assessments have been restricted to a limited number of models and data points. Clarifying the effects of these factors on performance scores can be more effectively achieved through a statistical lens. Our study embarks on a thorough re-examination of these LLMs, targeting the inadequacies in current evaluation methods. With the advent of a uniform evaluation framework, our research leverages an expansive dataset of evaluation results, introducing a comprehensive statistical methodology. This includes the application of ANOVA, Tukey HSD tests, GAMM, and clustering technique, offering a robust and transparent approach to deciphering LLM performance data. Contrary to prevailing findings, our results challenge assumptions about emergent abilities and the influence of given training types and architectures in LLMs. These findings furnish new perspectives on the characteristics, intrinsic nature, and developmental trends of LLMs. By providing straightforward and reliable methods to scrutinize and reassess LLM performance data, this study contributes a unique perspective on LLM training and core features.

## 1  Introduction

The advent of Large Language Models (LLMs) marks a significant milestone in the evolution of artificial intelligence. The advancement of LLMs has been remarkable, yet the foundational elements that govern their operations remain somewhat of a mystery. For instance, central to this puzzle is understanding why LLMs exhibit certain advanced abilities that their smaller counterparts do not (Wei et al., 2022). With the rapid emergence of numerous LLMs, it has become crucial to swiftly and effectively evaluate their performance adopting reliable and standardized approaches. The extraordinary pace at which LLMs are evolving presents challenges in fully grasping their nature, characteristics, and potentials. As mentioned above, the mystery issues on LLMs could potentially be resolved through efficient and thorough evaluations. To assess the effectiveness of LLMs, a significant number of tasks and benchmarks have been introduced (Chia et al., 2023; Liang et al., 2022; Zhao et al., 2023; Chang et al., 2023; Guo et al., 2023). Current evaluation datasets predominantly focus on specific abilities like language understanding, reasoning, and human alignment individually. Previous research identifies several critical measures that must be considered in the evaluation of LLMs, such as accuracy, efficiency, bias, safety etc. (Liang et al., 2022; Chang et al., 2023). Accuracy is paramount, encompassing not only factual correctness but also the precision of inferences and problem-solving. Consequently, current LLM evaluations tend to prioritize accuracy (Fu et al., 2023b; Safdari et al., 2023; Choi et al., 2023; Yuan et al., 2023; Li et al., 2023).

Recent evaluation efforts reveal several glaring issues. For instance, "emergent abilities" could be observed from a number of LLMs, such as GPT, PaLM and LaMDA (Wei et al., 2022; Schaeffer et al., 2023). Some researchers found that instruction-tuning provides a broad set of advantages compared with other types of training (fine-tune, pretrained, RL-tuned etc.)(Liang et al., 2022; Chung et al., 2022; Zhao et al., 2023). Zhao et al. (2023) also reported that the small-sized open-source models perform not well on mathematical

reasoning and scaling the open-source modes can improve the performance consistently. Researchers also found that some of the inconsistencies among the relationships between model size and task performance (Burnell et al., 2023). These findings actually are involved the overall performance of LLMs and different abilities with training types and scaling. However, the findings drawn from these studies primarily stem from observations made using a relatively small dataset. However, these findings have not undergone rigorous validation with a more extensive dataset. For enhanced reliability and accuracy of the results, further validation efforts could benefit from the application of comprehensive statistical methods. The following details these potential problems and challenges.

A primary issue is the narrow range of models typically assessed in multiple tasks — often several to 30 (Yu et al., 2023b; Yu et al., 2023a; Fu et al., 2023a; Jiang et al., 2023b; Huang et al., 2023), compared to the over 500000 models available, for instance, on `Huggingface`. Clearly, the proliferation of models has transformed them from isolated phenomena into a comprehensive ecosystem. Within this ecosystem, where numerous entities exist, it is essential to analyze the big data of the collective to understand their characteristics and developmental laws. However, the limited selection fails to capture the full spectrum of LLMs, diminishing our understanding of their features and diverse capabilities. For example, the limited number of LLMs may have emergent abilities. The question is whether these few models could represent the population of LLMs. As the array of LLMs expands and the availability of massive data on LLMs evaluation results, it is crucial to consider them collectively.Further, essential characteristics of LLMs, such as emergent abilities, scaling law Kaplan et al. (2020), should be analyzed using larger datasets and considering broader factors like training types and architectures, not just parameters like scale (e.g., parameter count, FLOPs). Moreover, the interplay of different LLM capabilities has not been examined. Understanding the potential interactions among their various abilities, similar to the interplay seen in human cognitive abilities (Conway et al., 2002; Buehner et al., 2006; Socher et al., 2022), remains a largely unexplored area in LLM research. Finally, the critical aspects of evaluating LLMs revolve around understanding the impact of scaling factors, training types, and architectural designs on their performance within the population of LLMs. This evaluation process bears resemblance to the assessment of human cognitive abilities, where factors like age, education, and sex are analyzed. These complex issues in both LLMs and human cognitive evaluations (or similar challenges in other fields) can be effectively addressed and validated through meticulous statistical testing and analysis.

A straightforward, reliable and efficient approach is essential to accurately assess the features and developing trends of LLMs. To achieve this, large-scale data on evaluation results is needed, using consistent evaluation datasets and standards across numerous LLMs. Fortunately, some researchers have begun establishing platforms for this unified data collection. Once collected, both basic and advanced statistical methods could be applied to thoroughly analyze these evaluation result data. When dealing with large-scale datasets on LLMs performance outcomes, we need to detach ourselves from the LLMs and not think about them. Currently, fundamental statistical techniques could be applied in the resulting data for understanding whether LLM performance varies significantly across different training types, architectures, and parameter sizes. Moreover, (non-)linear regression models can be employed to examine how training parameters or types affect LLM performance, and to explore the interactions among various LLM capabilities. These different statistical methods can cross-validate each other. These multifaceted statistical analyses will create a comprehensive framework, enabling an in-depth re-evaluation of the performance result data in LLMs, shown as in Fig.1.

## 2 Methods

It is prudent to establish a centralized platform that consolidates results from various LLMs, using identical datasets and evaluation criteria. The `Open LLM Leaderboard` serves this purpose, aiming to systematically track, rank, and assess LLMs `open-llm-leaderboard` (Available at `huggingface.co/spaces/HuggingFaceH4/open_llm_leaderboard`). This platform offers the extensive and cohesive dataset on LLM evaluation results. For our analysis, we extracted the data from the `Leaderboard`.

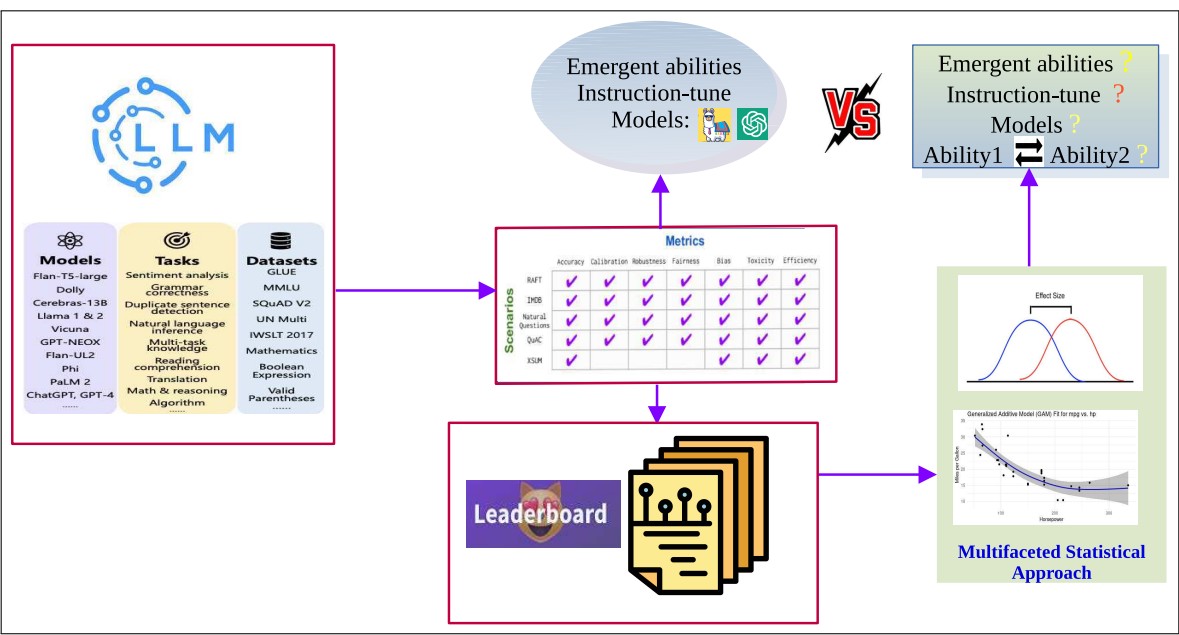

Figure 1: Reassessment methods for massive LLMs evaluation outcomes.

## 2.1 The datasets

The `Open LLM Leaderboard` evaluation process involves several benchmarks from *the Eleuther AI Harness* `eval-harness`, a unified framework for measuring the effectiveness of LLMs `open-llm-leaderboard`. Table 1 is a brief overview of each benchmark.

Table 1: The benchmark evaluation datasets in the `Open LLM Leaderboard`

| Name | Specificity | Capability | Scale | Number of Shots |
|------|-------------|------------|-------|-----------------|
| AI2 Reasoning Challenge (ARC) (Clark et al., 2018) | Grade-School Science Questions | Knowledge | 7787 science exam questions | 25-shot |
| HellaSwag (Zellers et al., 2019) | Commonsense Inference | Knowledge Reasoing -Complex Reasoning | 70000 questions | 10-shot |
| MMLU (Hendrycks et al., 2021) | Massive Multi-Task Language Understanding | Language Comprehension | 15908 questions, knowledge on 57 domains | 5-shot |
| Winogrande (Sakaguchi et al., 2019) | Adversarial Winograd Schema Challenge | Common Sense and Reasoning | 44k questions | 5-shot |
| GSM8k (Cobbe et al., 2021a) | Grade School Math Word Problems Solving | Symbolic Reasoning -Complex Reasoning | 8.5K math questions | 5-shot |
| TruthfulQA (Lin et al., 2022) | Propensity to Produce Falsehoods | Human Alignment | 817 questions | 0-shot |

These renowned and extensive evaluation datasets cover a wide range of capabilities including knowledge, complex reasoning, and human alignment. Knowledge utilization is tested through datasets like "ARC".Complex Reasoning is assessed via datasets like "HellaSwag", which emphasizes knowledge reasoning, and "GSM8k", centered on symbolic reasoning, with a focus on accuracy and problem-solving rates. For Human Alignment, "TruthfulQA" is used to gauge the truthfulness of responses from LLMs. Collectively, these benchmarks provide an assessment of a model's capabilities in terms of knowledge, reasoning, and some math, in various scenarios.The performance metric is recorded using scores, such as accuracy. These evaluation datasets are designed to assess multiple dimensions of cognitive ability in LLMs. The sample of the dataset is seen in Fig. 4 of Appendix A.

As of *January 12, 2024*, we collected the evaluation results on 1212 LLMs from the `Open LLM Leaderboard` which covered a broad spectrum of LLMs, ranging from open-source to closed-source API-accessible models. We excluded models with zero parameters, totaling unique **1186**. The dataset from the `Leaderboard` largely surpasses the size of most datasets on LLM evaluation results, which typically evaluate between a few and up

to 30 models, making it a more comprehensive resource for evaluation results. This number is expected to grow as more researchers contribute their results. The data on LLMs evaluation results from the `Leaderboard` also include a diverse range of factors: *architectures*, *training types*, and *hyperparameters*. Their (hyper)parameter counts span from $0.01B(billion)$ to $180B$, with an average ($\mu$) of $8.19B$, a median ($M$) of $6.61B$, and a standard deviation ($\sigma$) of $13.73B$. The training types are categorical factors, and they are categorized into five groups with their proportions: "fine-tune" (63%), "instruction-tune"(Wang et al., 2023) (15%), "pretrained" (13.5%), "RL-tune"(Reinforcement Learning from Human Feedback tune)(1.8%)(Zheng et al., 2023), and unknown (7%). Instruction-tuned models have been further refined with additional fine-tuning using various instructions, such as task datasets, daily conversations, and synthetic instructions. There are 31 distinct architectures (categorical factors), including "BloomForCausalLM", "GPT2LMHeadCustomModel", "LlamaForCausalLM", and others. These 31 architectures could be further classified into 12 broader categories with their proportions: "Bloom" (3%)(Workshop et al., 2022), "Falcon" (1%)(Almazrouei et al., 2023), "GLM" (1%)(Zeng et al., 2022), "GPT2" (9%)(Radford et al., 2019), "GPTJ" (3.3%)(Wang & Komatsuzaki, 2021), "GPTNeo"(11%)(Black et al., 2022), "Llama" (54%)(Touvron et al., 2023), "Mistral" (2.4%)(Jiang et al., 2023a), "OPT"(3.6%)(Zhang et al., 2022), "Rwkv"(1%)(Peng et al., 2023), and Other (9.9%). Additional factors, like "precision" and "Hub License", are also included in the data from the *Leaderboard*.

To ensure the robustness of our analysis, we cross-validated the other data on LLMs evaluation results, which is the supplementary dataset in the present study. This **supplementary dataset** comprises a comprehensive compilation of performance scores across several well-known LLM evaluation datasets, alongside detailed specifications of parameters and architectures. However, the supplementary dataset encompasses a relatively limited scope, covering **65** LLMs. We employed analogous re-evaluation methods to process and analyze it. The detailed outcomes of this supplementary dataset are reported in the **Appendix E**.

## 2.2 Re-evaluation methods

After gathering data from sources like the `Open LLM Leaderboard`, we applied a multi-faceted statistical approach to analyze the data. The following details the three statistical methods to re-evaluate the data on LLMs performance results.

**ANOVA and Tukey Tests**: We categorize data into groups based on factors like "architecture", "training types", and "parameter count", then apply ANOVA (Analysis of Variance) tests to identify significant differences across categories within each dataset. Where differences are significant, Tukey HSD (Honestly Significant Difference) tests are used for detailed pairwise comparisons. Such tests could help know whether data filtered by such factors really show significant differences on the data of performance scores. Specifically, we employed a two-step analytical method to compare differences across multiple categories within a dataset. First, we segments the data into subsets based on specific categories (such as architecture categories "Bloom", "GPT2", "GPTJ", etc., or training types "fine-tuned", "instruction-tuned", or parameter range scales, "0-1.5B", "1.5-3B" etc.) for various evaluation benchmark datasets, such as "ARC", "HellaSwag", and others. Each subset corresponds to a category within a given benchmark dataset. Then, we conduct ANOVA tests on these subsets to statistically evaluate the differences among the categories for each column. If the ANOVA results indicate significant differences, a post-hoc Tukey HSD test is applied for pairwise comparison between the categories. This method allows for a detailed understanding of how different categories influence the values in scores of the benchmark datasets, providing insights into the underlying patterns and relationships.

**GAMM**: Performance scores in one given evaluation dataset are treated as a **dependent variable**, while training parameters are considered an **independent variable**. Other factors, such as training types and architectures, are treated as **random variables**. Given this setup, we can use statistical regression models to carry out comprehensive tests and analyses, exploring the relationship between the dependent variable (performance scores) and the independent variable (training parameters). Generalized Additive Mixed Models (GAMM) can primarily analyze and model complex, non-linear relationships between dependent variable and independent variables in large datasets, which is implemented by the `mgcv` package in `R` (Wood, 2017; Wood & Wood, 2015). By employing smooth terms, the method adeptly captures non-linear relationships between the dependent variables (like scores of HellaSwag, ARC) and independent variables (Parameter count). This is crucial in real-world data where relationships are rarely strictly linear. The inclusion of random effects for variables (like architectures, training types) allows for the modeling of group-specific variations. The

method's application across different performance metrics (like HellaSwag, ARC, MMLU, etc.) demonstrates its versatility.

In short, GAMMs can capture the non-linear relationships between LLM performance and various predictors, such as model size, type of training, and architecture. By including random effects, we can account for variations between different LLMs and tasks. This means it can model how performance varies not merely based on fixed characteristics (like model size) but also due to inherent differences between individual models or datasets. GAMMs allow for simultaneously considering multiple predictors and their interactions, providing a more comprehensive view of what drives LLM performance.

**Clustering Analysis**: t-SNE (t-Distributed Stochastic Neighbor Embedding)(Van der Maaten & Hinton, 2008) effectively simplifies complex, high-dimensional data into a 2-dimensional space, making it easier to identify patterns, clusters, and relationships. This method is significant for its ability to transform and visualize complex datasets in a way that highlights underlying patterns and relationships. Overall, t-SNE aids in understanding how various variables (e.g., parameter, types) interact within the data clusters. This can also provide cross-verification with the results from the above two methods. (More details on statistical methods and data bias are seen in `Appendix D`.)

## 3 Results

### 3.1 Result 1: Difference analysis by parameter, and training type / architecture

We categorized the result dataset based on three distinct criteria: model training types, architectural frameworks, and parameter range scales. Notably, the model training types encompass five categories (fine-tuned, instruction-tuned, pretrained, RL-tuned, and unknown). The architecture models include 12 frameworks such as Bloom, Falcon, GLM, GPT2, GPTJ, GPTNeo, Llama, Mistral, OPT, Rwkv, and Others. The parameter ranges (billion) are segmented into distinct brackets, mirroring those used in the `Open LLM Leaderboard`: $[0, 1.5](23.1\%)$, $[1.5, 3](5.4\%)$, $[3, 7](39.3\%)$, $[7, 13](19.8\%)$, $[13, 35](10.8\%)$, and $[35, 80](2\%)$.

Our primary focus centers on the implications of parameter range scales. Employing ANOVA and Tukey's tests on scores from various benchmark datasets, we identified several significant findings (using a significance threshold defined by a $p$-value less than 0.05). In an analysis of various datasets, statistical significance varies across parameter ranges. In the six evaluation datasets, many comparisons between specific parameter ranges showed insignificant differences. In the "TruthfulQA" dataset, certain ranges exhibited significant differences, a contrast to the other datasets. It turned out aht the range **[3,7]** consistently demonstrated significant differences across multiple datasets. This indicates that only certain parameter scales are significant in the performance of LLMs. The detail is seen in the **Appendix A**.

Second, We conducted a detailed examination of model training types (fine-tune, instruction-tune, pretrained, RL-tune, and unknown categories). Throughout this analysis, we consistently applied the same significance threshold for comparability and rigor. We found that in the HellaSwag dataset, as well as in ARC, MMLU, TruthfulQA, and Winogrande, the differences between pretrained vs.fine-tuned and pretrained vs. instruction-tuned models are statistically significant. In the GSM8K dataset, the pretrained vs. instruction-tuned category stands out significantly. These results indicate a consistent, significant difference between pretrained and instruction-tuned models across all six evaluation datasets. However, no significant differences are noted between instruction-tuned and fine-tuned or RL-tuned models. This suggests that while **instruction-tuned** models demonstrate benefits compared to pretrained models, their advantages are **not** as pronounced when contrasted with **fine-tuned** models. In summary, the efficacy of instruction tuning is apparent, but it does not clearly surpass the benefits of fine-tuning.

Third, we categorized the data according to 11 architectures. Subsequently, ANOVA and Tukey's tests were conducted, applying a consistent threshold for the $p$-value across all analyses. Our analysis identified significant differences between pairs of architectures, as summarized in Table 2 in the **Appendix A**. This analysis reveals that architectures such as "GPT2", "Llama", "Bloom", "Mistral", and "GPTNeo" are particularly effective in executing evaluation tasks. Their recurrent appearance in significant pairings underscores their robust performance across different benchmarks.

Additionally, the correlations among the scores of the six evaluation datasets are illustrated in Fig.5 in the **Appendix A**. The correlationship reveals that "TruthfulQA", representing human alignment, does not exhibit a high correlation with the abilities represented by the other evaluation datasets.

### 3.2 Result 2: GAMM analysis: Emergent abilities and interplay of various abilities

Generally "emergent abilities" in LLMs are unexpected capabilities that suddenly appear in larger models, absent in smaller ones, representing a qualitative leap in performance that often resembles human-like reasoning and cannot be predicted by simply extrapolating from smaller models. The related research has identified the two defining properties of "emergent abilities" in LLMs: a). Sharpness, transitioning seemingly instantaneously from not present to present; b) Unpredictability, transitioning at seemingly unforeseeable model scales (Schaeffer et al., 2023). Actually, emergent abilities hold by default that c) such abilities do not decrease as the training size becomes larger. Schaeffer et al. (2023) took various metrics to scrutinize the purported emergent abilities, shown as in Fig. 2, discovering inconsistencies between these abilities and established principles.

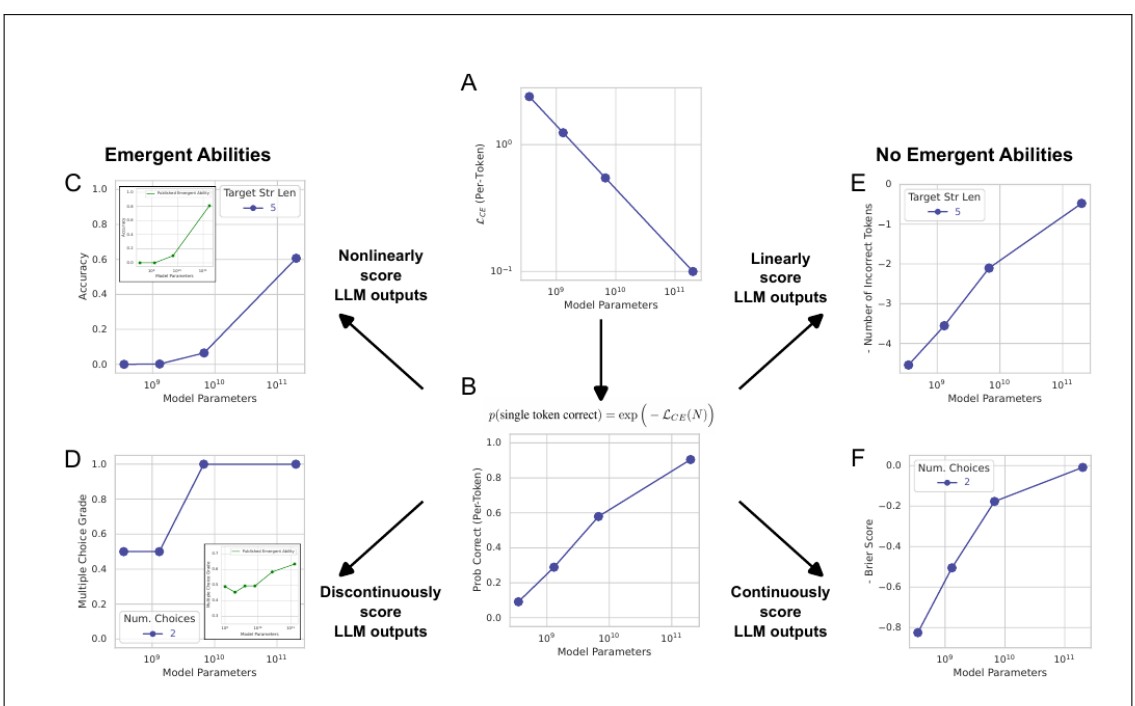

Figure 2: Emergent abilities of LLMs are created by the chosen metrics (Schaeffer et al., 2023), not unpredictable changes in model behavior with scale.

The appropriateness of employing various metrics to transform raw data on evaluation results remains uncertain. Nevertheless, more straightforward and dependable statistical methods can be employed for verification. From a statistical analysis standpoint, performance scores are considered as the dependent variable, with training parameters acting as independent variables. Other factors, such as training types and architecture, are regarded as random variables. This arrangement allows for the application of advanced regression models (e.g., GAMM) to conduct detailed testing and analysis.

To elucidate the relationship among variables, we executed multiple sets of GAMM fittings. The first group aims to comprehend the impact of parameter count on performance scores while considering random effects. For this purpose, the specific GAMM equation employed is as follows:$(log\_HellaSwag \sim s(log\_Param) + s(Architecture, bs = "re"), data = data)$. Here $\mathtt{s}$ is smooth, $\mathtt{re}$ = random effect. The $\mathtt{s}$ in a GAMM model represents the smooth functions applied to predictors, allowing for flexible modeling of non-linear relationships in data, while also incorporating random effects to handle correlated groups or clusters within the data. In

other words, the non-linear relationship in the data could be better detected using the smooth function. we applied a logarithmic transformation to the variables, which brought the data closer to a normal distribution. This transformation enables us to achieve more accurate fittings using GAMM. Furthermore, the addition of the random variable "Type" in the equation did not show significant results. This lack of significance could be attributed to the overshadowing effect of the "Architecture" factor, and "Architectures" is strongly significant across the evaluation datasets. The results are shown in Fig. 3, where the parameter has a strongly significant effect on each of evaluation datasets (using a significance threshold defined by a *p*-value less than 0.001).

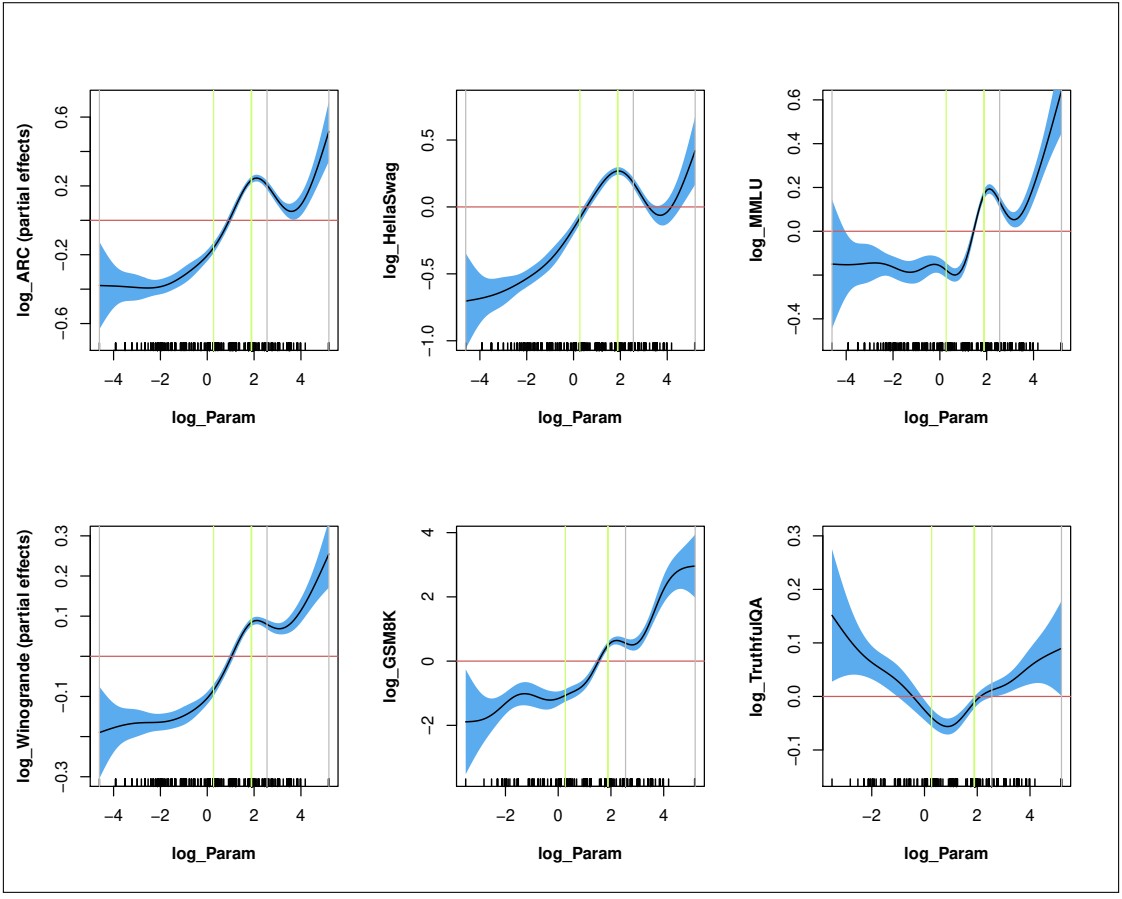

Figure 3: Partial effects of parameters on LLMs performance scores. (In each plot, the vertical lines signify five quartiles, which divide the data on parameters into five equal quartiles: 0%[0.01B] - 20%[1.31B] - 40%[6.53B] - 60%[6.65B] - 80%[12.85B] - 100%[180B]. *X-axis* "log_Param" is the logarithm of model training parameters, and *y-axis* is the logarithm of scores in each evaluation dataset. A curve of partial effects represents the relationship between a predictor variable and the response variable. Steeper slopes suggest a stronger relationship, and flatter slopes imply a weaker one. A curve of partial effects represents the relationship between a predictor variable and the response variable when a curve fluctuates around zero, it indicates that the effect is weak. The pointwise 95%-confidence intervals are shown by the blue shadow.)

Referencing the standards illustrated in Fig. 2 and the descriptions by Wei et al. (2022), we observed discrepancies between the emergent abilities depicted and the findings in our Fig.3. Specifically, Panels E and F in Fig. 2 lack flattened stages in their curves, which contradicts the principle of sharpness. Similarly, the curves in the six plots in Fig. 3 also show no flattened stages, aligning with the situation in Panels E and F of Fig. 2. However, while the abilities in Panels E and F of Fig.2 continually increase, the curves in Fig.3 exhibit unpredictable changes between the 60% and 100% quartiles (6.65B-12.85B-180B). For ARC, HellaSwag, and MMLU, the curves exhibit a wave-like, non-linear fluctuation. In the first two cases, abilities at the final stage do not increase significantly post-fluctuation, leveling near the 60%-80% quartile range.

MMLU shows no increase until the 20% quartile. Winogrande and GMS8K, in contrast, display largely linear curves. TruthfulQA stands out with a unique curve pattern: a decrease at the initial stage, an increase between the 20% and 80% quartiles, yet a final level lower than the initial one, with unpredictability post-80% quartile. Fig.3 confirms that the parameter range [**3,7**], as discussed in section 3.1 and analyzed using ANOVA and Tukey's method, demonstrates strong significance across multiple abilities. This parameter range consistently enhances scores across the evaluation datasets, as illustrated in Fig. 3. From Fig.3, the trends are partly consistent with "emergent abilities" descriptions. Notably, there is no initial stage flattening, implying the abilities do not emerge suddenly. However, from the 60% quartile, the four plots in Fig. 3 show some unpredictable tendencies, which could align with emergent abilities as described in the literature. It seems that "GMS8K" and "Winogrande" keep increasing with larger parameter scales. If "emergent abilities" imply an increasing trend with larger parameter sizes, then TruthfulQA's performance contradicts this, suggesting that such abilities might be present in certain parameter size ranges but not uniformly across all stages.

Additionally, due to the uneven proportion of some architectures (e.g., Llama-54%, GPTNeo-11%, GPT2-9%, Bloom-3% etc.), we scaled the data on the scores in different evaluation datasets in order to mitigate such unbalanced data. Based on the architecture categories, we calculated scaling factors to normalize the distribution across these categories. It then applied these scaling factors to various performance metrics in the dataframe. This kind of processing is useful in situations where models are fairly across different categories that might be unevenly represented in data. Using the same GAMM fittings, we can get the similar results as the data are not scaled. The visualization of the results is shown in Fig.8, Fig.9, and Fig. 10 in the **Appendix B**.

Our interest extends to exploring how different training types (such as fine-tune, instruction-tune, etc.) might influence the changes in abilities. To investigate this, we incorporated additional random effects into our analysis, yielding some notable observations. Specifically, we introduced the training type as a grouping factor in the GAMM fitting: $(log\_HellaSwag \sim s(log\_Param, by = Type) + Type + s(Architecture, bs = "re"), data = data)$. "$s(log\_Param, by = Type)$" specifies a smooth function (**s**) of the logarithm of parameter, with a different smooth for each level of the "Type" variable. This allows for non-linear relationships between "log_Param" and "log_HellaSwag" that differ depending on the "Type", and it suggests that 'Type' is included as a categorical predictor in the model. The visualization is seen Fig. 7 in the **Appendix B**. Our analysis revealed that the six evaluated abilities demonstrate significant results across both fine-tuning and instruction-tuning levels. However, in the context of RL-tuning, certain cases were identified as significant, characterized by a $p$-value exceeding 0.05. In the scenario involving pretrained models, specifically for MMLU, the observed curve predominantly fluctuated around zero. These observations align coherently with the findings derived from ANOVA and Tukey tests, as detailed in Section 3.1.

The subsequent analysis delves into the interplay among various abilities. These models possess a range of capabilities, including language understanding, commonsense reasoning, and mathematical reasoning, which may interact and influence each other. To comprehend how a certain ability affects other abilities, we approached the analysis by considering the particular ability as the dependent variable, while treating the other capabilities as independent factors. This allows us to explore the potential impact some abilities may have on one given ability. To acheive this, we employed the following GAMM equation: $(log\_ARC \sim s(log\_HellaSwag) + s(log\_MMLU) + s(log\_Winogrande) + s(log\_GMS8K) + s(log\_TruthfulQA) + s(Architecture, bs = "re"), data = data)$. For each ability, we employed a similar GAMM equation for estimation with random effects. When focusing on a particular ability as the dependent variable, that specific ability was excluded from the independent factors. The findings from this approach are presented in Fig.6 in the **Appendix B**. Our analysis revealed that a given ability influence other capabilities, while others do not certainly exert a significant impact (detailed in the **Appendix B**). To summarize, the abilities represented by "HellaSwag" and "MMLU" showed significant effects on other abilities, suggesting that knowledge reasoning and language understanding might play a pivotal role in influencing LLMs' overall capabilities. Specifically, both "ARC" and "HellaSwag" demonstrated a general influence across various abilities, as did "MMLU". In contrast, the remaining three abilities displayed insignificant effects (with $p$-values greater than 0.05), indicating a lack of a general impact on the other capabilities.

Additionally, the similar AVNOVA, Tukey and GAMM tests were done in the supplementary dataset, and the results are basically consistent with the ones in the primary dataset, The details are seen the **Appendix E**.

### 3.3 Clusters with key factors

We employed t-SNE to compress data and facilitate clustering. The method's findings could check with the results from ANOVA and GAMM tests to some degree. For instance, certain parameter ranges did not form distinct clusters, and the clustering based on training types also lacked clear differentiation. The cluster results is shown in Fig.11 in the **Appendix C**.

In Fig.11, the first left panel indicates that the parameter range of [1.5,7] (in blue) forms a cluster. Conversely, the range of [0.01,1.5] forms a smaller cluster. Other parameter ranges, however, do not exhibit such clear clustering. The middle panel suggests that while fine-tuned models cluster to some extent, they are interspersed by instruction-tuned models. In the right panel, the Llama architecture appears to create a cluster. Despite the formation of clusters, they are not distinctly separated from one another. This observation implies that clusters formed based on certain factors lack strong significance. It also suggests that the impact of specific parameter ranges, instruction-tuning, or certain architectures may not be as influential as anticipated. These clustering patterns, however, offer valuable cross-verification evidence that aligns with the results obtained from ANOVA and GAMM tests, contributing to a more comprehensive understanding of the data.

## 4 Discussion

We employed multiple statistical techniques, including ANOVA, Tukey HSD tests, GAMMs, and clustering analysis to explore the impact of scaling factors, training types, and architectural designs on LLM performance, interactions among various LLM capabilities. We also validated previous findings on emergent abilities and the influence of training types and architectures. This section provides a detailed account of these findings and makes further analysis.

### 4.1 Non-linear scaling and target scaling

Our study presents findings that challenge certain established conclusions regarding the evaluation of LLMs in previous research. The previous research could be summarized as " linear scaling". However, we found non-linear scaling in LLM development(Hoscilowicz et al.). The following details these findings.

First, we question the purported superiority of instruction-tuning over fine-tuning. While previous studies (Wei et al., 2021; Liang et al., 2022; Zhao et al., 2023) indicate that instruction-tuned models generally outperform base models, our data does not support this assertion. According to our ANOVA and Tukey tests, no significant differences were observed between instruction-tune, fine-tune, and RL-tune across six evaluation datasets. Second, regarding the performance of small-sized, open-source models in mathematical reasoning, Zhao et al. (2023) reported their underperformance. However, referring to Fig. 3, Fig. 8, Fig. 8, and Fig. **??**, if we define the parameter range from the first to the third quartile as small-sized, these models exhibit comparable performance in mathematical reasoning tasks (e.g., GMS8K) to their larger-scaled counterparts. Third, Zhao et al. (2023) claim that the 'Llama' model outperforms others is not corroborated by our analysis. We found that 'Llama', along with other models like 'GPT2', 'Minstral', and 'Falcon', show equivalent proficiency in complex reasoning tasks (GMS8K and ARC). Fourth, while Zhao et al. (2023) suggested that scaling up open-source models consistently enhances performance, our findings indicate that this may be task-dependent.

Moreover, as parameter sizes grow much larger, their effects become unpredictable. This suggests that scaling up models within a certain range can consistently improve performance, but beyond that range, the outcomes become uncertain.

Further, our data supports a shift from uniform scaling to targeted scaling in LLM performance, emphasizing the effectiveness of increasing model capacity in areas most relevant to desired capabilities. By focusing on task-specific improvements and adaptive compute allocation based on prompt difficulty, targeted scaling offers a more nuanced and efficient path to enhancing LLM capabilities (Snell et al., 2024). This strategy not only improves performance in key areas but also allows for more cost-effective and adaptable model development,

potentially revolutionizing how we approach the scaling and optimization of large language models in the future.

To summarize, the development of LLMs is characterized by **non-linear scaling** and **target scaling**, with diminishing returns as models grow larger. The relationship between size and capability varies by task. **Architectural diversity** is key, as different structures excel in specific areas. No single architecture dominates all benchmarks. **Training methods also matter**. Instruction-tuning is not always superior, with fine-tuned and pretrained models remaining competitive in certain domains.

### 4.2 Gradual emergent abilities and task-specific emergence

Next, we discuss emergent abilities. As discussed in the section on Result 2, previous related research identified three characteristics of emergent abilities in large language models (LLMs): a) Sharpness, b) Unpredictability, and c) Continuous improvement (Schaeffer et al., 2023). The emergence of some capabilities in LLMs might be attributed to their training, as inferred from comparisons with smaller-sized models. However, the presence of certain abilities in the majority of LLMs does not necessarily imply that these abilities are intrinsic characteristics.

As illustrated in Fig. 3, these abilities manifest even with minimal parameters (as low as 0.01 billion). Our research indicates a consistent increase in the capabilities of the models from the outset up to the point where 60% of the data is utilized, aligning with the observations reported in Schaeffer et al. (2023). However, the aspect of unpredictability, which our study identifies beyond the 60% data threshold, was not observed or reported in the findings of Schaeffer et al. (2023). However, "GMS8K" and "Winogrande" seems to keep increasing, which are consistent with some research (Cobbe et al., 2021b; Chowdhery et al., 2023; Touvron et al., 2023). It is possible that mathematical reasoning ability does not show unpredictable tendency when the parameter is up to 180B.

In the current study, a degree of unpredictability is evident when the parameter size exceeds 7B. In essence, the abilities of LLMs tend to scale almost linearly with parameter sizes up to 7B. Beyond this threshold, their performance becomes more predictable (Ganguli et al., 2022). The applicability of the scaling law appears to be confined within a specific range and may also vary depending on the nature of the tasks involved (Kaplan et al., 2020). Additionally, the emergence of some abilities is also task-dependent. In short, the abilities of LLMs continue to improve with increasing training sizes, but performance becomes less predictable as the training size reaches extremely large scales. All this indicates that simply expanding the training size may not be a consistently reliable method for enhancing all capabilities of LLMs (Patel, 2024).(More is seen in **Appendix D**.)

To conclude, our findings challenge the concept of "emergent abilities". We observe "**gradual emergent abilities**", suggesting a more continuous development of capabilities that may be interrupted or even regress at times. This process is characterized by gradual, task-dependent, and potentially non-linear progress rather than abrupt breakthroughs (**?**). Interestingly, as of 2024, the popularity of training giant LLMs has somewhat declined. This trend may reflect concerns about the potential superior power of such models, aligning with our research on gradual capability development.

### 4.3 The interaction of abilities

In the following, we focus on the interplay among various abilities in LLMs. Fig. 6 suggests that knowledge reasoning and language understanding could have an overall impact on the other capabilities of LLMs. Burnell et al. (2023) used the methods of factor analysis and correlation to identify three essential capabilities based on HELM (Liang et al., 2022): language comprehension, language modeling, and reasoning, and their finding is basically consistent with our findings. However, our study did not include the data on language modelling (e.g., word prediction, text generation).

Our findings suggest that different capabilities may develop at varying rates and in different ways as models scale or undergo different types of training. Our data also reveals that performance improvements varied across different tasks and benchmarks, indicating that capabilities interact differently depending on the specific task. Moreover, architectural differences had a significant impact on how different capabilities interacted

and developed across various benchmarks. The interactions between capabilities were also influenced by the type of training (e.g., fine-tuning, instruction-tuning, pretraining), suggesting a complex relationship between training methodology and capability development.

In short, LLM abilities exhibit **complex interplay**. Some fundamental capabilities may have broad effects on other abilities. Non-uniform development and task-Dependent performance were observed in the ability interplay. The interplay effects were also impacted by architectural differences, type of training etc. Understanding these capability interactions could inform more targeted approaches to LLM development, potentially allowing for the optimization of specific capabilities without compromising others.

### 4.4   Training strategies

Our findings suggest revising model training strategies for LLMs. First, we recommend a cautious scaling strategy, supported by diverse metrics, due to unpredictability at large scales. Specifically, as model training parameters increase, so does their riskiness. This is particularly crucial for individuals and small firms to consider, as larger scales do not necessarily yield better results, and must be weighed against the cost of inputs and potential risks. Moreover, a model's performance is determined by numerous factors. Ultimately, the non-linear relationship between training size and effectiveness beyond 7 billion parameters presents significant challenges. Second, our results indicate that current instruction-tuning methods could be less effective than expected, highlighting the need for further improvements. Third, imbalances in various LLM capabilities appear more significant than assumed, suggesting that enhancing certain abilities could benefit general-purpose models. Certain architectural choices may be more beneficial for specific types of tasks, challenging the notion of a one-size-fits-all approach to LLM design, that is, architecture-task alignment (Achintalwar et al., 2024). Data Our findings also suggest that the quality and diversity of training data may be more crucial than sheer quantity in developing certain LLM capabilities. Overall, these insights help refine training strategies and deepen our understanding of LLM core features and their interrelationships.

Finally, we address the disparities between open-source and closed-source LLMs. The `Open LLM Leaderboard` predominantly features a limited selection of well-known closed-source LLMs such as GPT-4, Gopher, LaMDA, PanGu, and Anthropic, despite the vast array of over 500,000 open-source models hosted on `Huggingface`. Evaluating closed-source LLMs, which are typically not publicly accessible, on comprehensive datasets is expensive and presents a major challenge for researchers. Nevertheless, including additional data from these closed-source LLMs in our analysis is unlikely to significantly impact overall statistical outcomes. Some studies prioritize these closed-source models, claiming they represent LLMs universally. However, this stance could be contentious considering the broader accessibility and prevalence of open-source models. These findings are derived from analyzing LLMs as a collective ecosystem using large-scale, standardized evaluation datasets. These characteristics are observed from an ecosystem or population perspective, rather than focusing on a single model.

## 5   Conclusion

The present study provides a more robust and transparent understanding of LLM performance, addressing limitations in previous evaluations that were often based on smaller datasets and fewer models. It challenges assumptions about emergent abilities and the influence of certain training types and architectures in LLMs. This study challenges the common belief that increasing model size is the primary means to boost AI performance, advocating for a unique approach grounded in rigorous multifaceted statistical analysis. It reveals that merely changing model size or architecture might not guarantee better outcomes, potentially impacting efficient resource use in real-world model training. The study underscores the complex interactions within various abilities of LLMs, contributing to a broader understanding that could influence public policy and educational programs, fostering a more informed interaction with AI technologies and AGI. This study provides new perspectives on the characteristics, intrinsic nature, and developmental solutions of LLMs, contributing a unique perspective on LLM training and core features. Conclusively, it calls for a holistic and meticulous evaluation of LLMs, with implications for academic and societal advancement, emphasizing thoughtful integration of these technologies into societal frameworks.

However, several limitations exist in the current research. The `Open LLM Leaderboard` is rapidly expanding, with recent additions like Llama3 and Grok leading to more finely-tuned models. We anticipate including the results testing on these new models and their fine-tuned models in future updates of the `leaderboard`. Additionally, the `leaderboard` currently lacks other training parameters. Providing more of various parameters could enable deeper and more insightful analysis. Another issue is the inconsistent use of datasets across different strategies; for example, instruction-tuning and fine-tuning often employ different datasets. It is impractical to standardize datasets across models and tuning strategies because different strategies determine to use different datasets. It is not convincing to argue that this inconsistency can affect the comparability of results. If some given strategy is really useful, the method should be effective for different models rather than merely only few models.

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

## Appendix

### A. ANOVA, Tukey tests and correlations

Table 2: The pairs and their respective frequencies of significance using Tukey tests on the score data considering the architecture types

| Model pairs | Frequency |
| --- | --- |
| OPT-Llama | 5 |
| Llama-GPT2 | 5 |
| Llama-Bloom | 4 |
| Mistral-GPT2 | 4 |
| Other-Llama | 4 |
| Llama-GPTNeo | 4 |
| GPTNeo-GPT2 | 3 |
| Other-GPT2 | 3 |
| GPTJ-GPT2 | 3 |
| Other-GPTNeo | 3 |
| Rwkv-Llama | 3 |
| Mistral-Bloom | 2 |
| Llama-Falcon | 2 |
| OPT-GPT2 | 2 |
| OPT-Mistral | 2 |
| Llama-GLM | 2 |
| Mistral-GPTNeo | 2 |
| Llama-GPTJ | 2 |
| Mistral-GPTJ | 1 |
| Rwkv-Mistral | 1 |
| Mistral-Llama | 1 |
| Mistral-GLM | 1 |
| Mistral-Falcon | 1 |
| GPTNeo-GPTJ | 1 |
| OPT-GPTNeo | 1 |

The following details the significant parameter range scales on various evaluation datasets. Specifically, when evaluating the HellaSwag dataset, we observe that the differences between the [13,35] and [1.5,3] ranges, as well as between [7,13] and [3,7], are statistically insignificant. Similarly, in the ARC dataset, the difference between the [7,13] and [3,7) ranges is not significant. In the case of MMLU, the parameter ranges [1.5,3] and [0,1.5], [3,7] and [13,35], [7,13] and [3,7] are found to be insignificant. However, for TruthfulQA, the ranges [1.5,3] and [0,1.5] and [7,13] and [1.5,3] demonstrate significant differences, while other range comparisons do not yield significant results. In the Winogrande dataset, the comparisons between [3,7] and [13,35], [7,13] and [13,35], and [7,13] and [3,7] are not significant. Finally, in the GMS8K evaluation, the comparisons of [1.5,3) and [0,1.5], [3,7] and [13,35], [7,13] and [13,35], [7,13] and [3,7] do not exhibit significant differences. Intriguingly, the [3,7] range consistently shows significant differences when compared with certain other ranges across these datasets. This analysis underscores the nuanced relationship between parameter ranges and dataset performance, revealing critical insights into the varied impacts of these parameters across different benchmarks.

The following details the tests on architecture types. Table 2 lists the pairs and their respective frequencies of significance using Tukey tests on score data on various architectures. Table 2 shows that the frequent occurrence of these models in significant pairings across multiple benchmarks highlights their consistent and robust performance. This analysis demonstrates that certain architectures, specifically 'GPT2', 'Llama', 'Bloom', 'Mistral', and 'GPTNeo', exhibit exceptional effectiveness in performing various evaluation tasks.

Fig. 4 is a sample from the dataset from the `Open LLM Leaderboard`. The variables in the dataset include training type, architecture, parameters, scores in different evaluation datasets. Fig 5 illustrates the correlations among several abilities represented by their evaluation datasets.

Moreover, the result we reported in the main body did not consider training scale for these types or architectures, that is, our analysis using ANOVA and Tukey tests only examined if there is any significant difference between different training types / model architectures and their performance scores. Moreover, we have applied the scaling method to normalize the performance data in GAMM analysis. The factor of training scale has been applied in the data on abilities scores (i.e. using the scaling method to normalize the score data). However, there is no difference between the raw data and the normalized data.

| | Model | Average | ARC | HellaSwag | MMLU | TruthfulQA | Winogrande | GSM8K | Type | Architecture | #Params (B) |
|---|---|---|---|---|---|---|---|---|---|---|---|
| | 01-ai/Yi-34B | 69.42 | 64.59 | 85.69 | 76.35 | 56.23 | 83.03 | 50.64 | pretrained | YiForCausalLM | 34 |
| | 01-ai/Yi-34B-200K | 70.81 | 65.36 | 85.58 | 76.06 | 53.64 | 82.56 | 61.64 | pretrained | LlamaForCausalLM | 34.39 |
| ○ | 01-ai/Yi-34B-Chat | 65.32 | 65.44 | 84.16 | 74.9 | 55.37 | 80.11 | 31.92 | instruction-tuned | LlamaForCausalLM | 34.39 |
| ○ | 01-ai/Yi-34B-Chat | 63.17 | 65.1 | 84.08 | 74.87 | 55.41 | 79.79 | 19.79 | instruction-tuned | LlamaForCausalLM | 34.39 |
| | 01-ai/Yi-6B | 54.08 | 55.55 | 76.57 | 64.11 | 41.96 | 74.19 | 12.13 | pretrained | LlamaForCausalLM | 6.06 |
| | 01-ai/Yi-6B | 54.02 | 55.55 | 76.42 | 63.85 | 41.86 | 73.8 | 12.66 | pretrained | YiForCausalLM | 6 |
| | 01-ai/Yi-6B-200K | 56.76 | 53.75 | 75.57 | 64.65 | 41.56 | 73.64 | 31.39 | pretrained | LlamaForCausalLM | 6.06 |
| | 01-ai/Yi-6B-200K | 56.69 | 53.58 | 75.58 | 64.65 | 41.74 | 74.27 | 30.33 | pretrained | LlamaForCausalLM | 6.06 |
| ○ | 0x7194633/fialka-13B-v3 | 34.58 | 30.97 | 48.83 | 26.36 | 40.58 | 59.43 | 1.29 | instruction-tuned | GPT2LMHeadModel | 12.85 |
| ○ | 0x7194633/fialka-13B-v3.1 | 34.11 | 29.95 | 47.28 | 25.41 | 43.03 | 58.48 | 0.53 | instruction-tuned | GPT2LMHeadModel | 12.85 |
| ○ | 0x7194633/fialka-7B-v3 | 46.4 | 48.55 | 71.05 | 43.06 | 44.79 | 69.46 | 1.52 | instruction-tuned | LlamaForCausalLM | 6.74 |
| ◆ | 42dot/42dot_LLM-PLM-1.3B | 35.7 | 32.42 | 56.39 | 27.09 | 38.68 | 58.88 | 0.76 | fine-tuned | LlamaForCausalLM | 1.44 |
| ○ | 42dot/42dot_LLM-SFT-1.3B | 36.61 | 36.09 | 58.96 | 25.51 | 39.98 | 58.41 | 0.68 | instruction-tuned | LlamaForCausalLM | 1.44 |
| ◆ | 64bits/LexPodLM-13B | 51.14 | 57.76 | 81.04 | 48.38 | 43.48 | 76.16 | 0 | fine-tuned | LlamaForCausalLM | 12.85 |
| | AI-Sweden-Models/gpt-sw3-126m | 28.49 | 22.18 | 29.54 | 24.43 | 44.03 | 50.67 | 0.08 | pretrained | GPT2LMHeadModel | 0.19 |
| | AI-Sweden-Models/gpt-sw3-126m | 28.45 | 22.01 | 29.56 | 24.53 | 44.07 | 50.43 | 0.08 | pretrained | GPT2LMHeadModel | 0.19 |
| ○ | AI-Sweden-Models/gpt-sw3-126m-instruct | 28.2 | 23.38 | 29.88 | 23.78 | 42.65 | 48.54 | 0.99 | instruction-tuned | GPT2LMHeadModel | 0.19 |
| | AI-Sweden-Models/gpt-sw3-20b | 40.71 | 41.81 | 68.75 | 28.47 | 37.1 | 67.17 | 0.99 | pretrained | GPT2LMHeadModel | 20.92 |
| ○ | AI-Sweden-Models/gpt-sw3-20b-instruct | 43.7 | 43.17 | 71.09 | 31.32 | 41.02 | 66.77 | 8.79 | instruction-tuned | GPT2LMHeadModel | 20.92 |
| | AI-Sweden-Models/gpt-sw3-356m | 30.41 | 23.63 | 37.05 | 25.93 | 42.55 | 53.04 | 0.23 | pretrained | GPT2LMHeadModel | 0.47 |

Figure 4: The sample of the dataset from the `Open LLM Leaderboard`.

## B. GAMMs and the partial effects on scaled Data

First, let's delve into the workings of Generalized Additive Mixed Models (GAMMs). At their core, GAMMs operate based on the following fundamental mathematical equation:

$Y = \beta_0 + f_1(X_1) + f_2(X_2) + \ldots + f_n(X_n) + Zb + \epsilon$ $Y$ is the response variable. $\beta_0$ represents the intercept.

$f_i(X_i)$ are the smooth functions of the predictor variables $X_1$, $X_2$, $\ldots$, $X_n$X $Z$ is the design matrix for

the random effects. $b$ symbolizes the random effects. $\epsilon$ is the error term, typically assumed to be normally

distributed. Each $f_i$ is modeled using basis functions like splines, with complexity controlled by smoothing parameters. The random effects $Zb$ are assumed to follow a normal distribution, often with zero mean and a specific covariance structure.

This approach allows for a more intricate analysis of data relationships, as it combines multiple smooth functions with parametric elements. Thus, GAMMs are designed to discern a non-linear effect only when the data robustly suggests such a pattern. Conversely, it will identify a linear effect when the data predominantly supports a linear relationship.

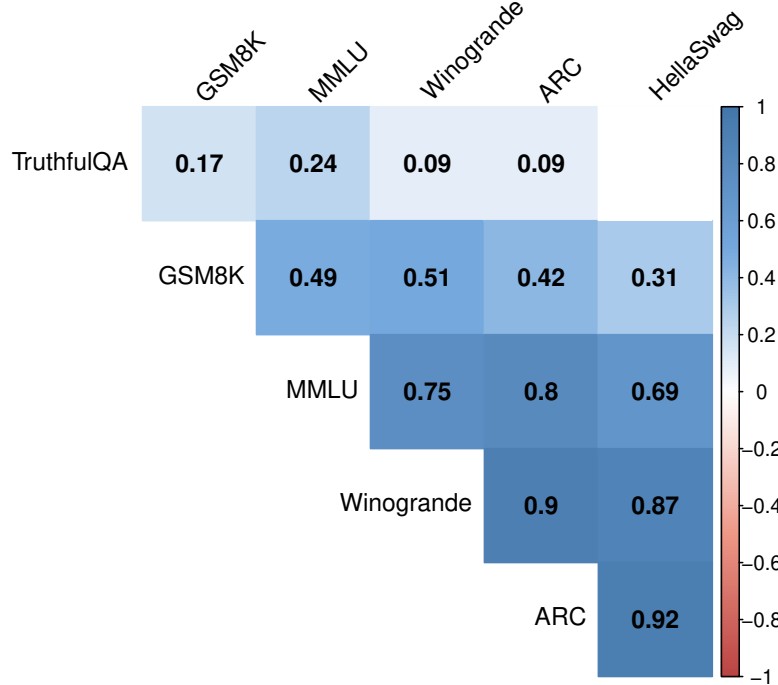

Figure 5: Correlations among various benchmark datasets

The following is the demonstration of visualization of GAMM analysis. First, the visualization of one given ability on other abilities in LLMs is illustrated in Fig.6. Fig. 7 shows how the parameter scale takes effect on the various abilities through different training types. As Fig. 7 shown, several insignificant cases can be seen (i.e., $p$-value is greater than 0.01). In our analysis, one particular instance within the RH-tuned model, when evaluated with "TruthfulQA", appears to be statistically insignificant. Additionally, the significance of the other two instances in the RH-tuned context is not particularly robust. This observation might suggest that the performance of the RH-tuned model has not meet the expectations. When examining the overall patterns exhibited in the fine-tune, instruction-tune, and pretrained models, we observe that their general curve shapes bear resemblance to each other. However, the specific abilities of these models demonstrate slight variations across the different training methodologies. Our findings from the GAMM analysis largely corroborate with the insights obtained from the ANOVA and Tukey tests, particularly concerning the distinctions among various training types. This consistency across different statistical methods reinforces the reliability of our results, highlighting nuanced differences in model performance based on the training approach.

Next, we visualized the GAMM results using the scaled data. Fig. 8 show how the parameter scale takes effect on the various abilities based on the scaled data which are normalized accordance to the frequency of architectures. Fig. 8 are consistent with Fig. 3.

We would like to delve deeper into the analysis of the "TruthfulQA" plot. The data shows a notable pattern based on the parameter range. Specifically, when the parameter is below the third quartile (i.e., 5.84B), there is a decrease in the score as the parameter size increases. In contrast, beyond the 5.84 B threshold, an improvement in performance is observed. Yet, this trend reverses with extremely large parameters (potentially exceeding 130B), where the score declines again. It is important to note that the peak scores, observed in the fourth and fifth quartiles, still fall short of the levels seen with smaller parameters. This pattern in

"TruthfulQA" finds echoes in other research. For instance, Fig. 4 in Lin et al. (2021) illustrates a similar trend in the 'Average Truthfulness' of generation tasks across five models, showing a decline in scores with larger model parameters. This phenomenon is also evident in the multiple-choice tasks of the same study. Moreover, Zhao et al. (2023) observed in their "TruthfulQA" evaluations that scores did not uniformly increase with larger model parameters. Such trends are not limited to "TruthfulQA" but seem prevalent in other Human Alignment tasks, such as C-Pair, WinoGender, RTP, HaluEVa. Based on these observations, it is hypothesized that other human alignment tasks might exhibit similar patterns to those demonstrated in "TruthfulQA" in our current analysis.

Using the same scaling data, we plotted how the parameter scale takes effect on the various abilities through different training types, as shown in Fig. 9. Fig. 9 is basically consistent with Fig. 7.

Meanwhile, using the same scaling data, we plotted how the various abilities influence with each other, as shown in Fig. 10.

The T-sne clusters are shown in Fig. 11. The clusters of those claimed factors are not clearly separated, which are also consistent with the findings using Tukey and GAMM tests.

## C. T-sne clusters

## D. The issues on statistical models, data bias and resulting insights

Our study applies statistical methods to evaluate LLMs, a practice not yet widespread in deep and machine learning. We assess the impact of scaling, training types, and architecture on LLM performance, similar to evaluating human cognitive abilities and other similar issues. There is a big trap to evaluate the performance of LLMs. Specifically, professionals in this field often focus intently on performance-enhancing data regarding LLMs. However, it is important to recognize that such data can be altered or even manipulated to some extent. When analyzing large-scale datasets on LLM performance outcomes, it is essential to maintain objectivity and avoid preconceptions about the LLMs themselves. Instead, we merely scrutinize these data objectively and without bias. Only then can we truly discern inherent features and laws about LLMs and their development from these data on their performance.

These analyses utilize tools like ANOVA, Tukey Tests, and GAMMs, directly applied to datasets without the need of training models. Our multi-faced statistical analysis supports the objective and honest approach to the data, sourced from the `Open LLM Leaderboard`, despite certain limitations due to unavailable factors. Nevertheless, our research offers new insights, underlining that the absence of some data does not detract from our methods' validity or our findings' significance. For instance, If the REHF is useful, the method should be useful for different models.

Our methodology was designed to evaluate general trends across the board, rather than focusing on isolated instances. This involved comparing the different techniques (e.g, REHF, fine-tuning) across various models, even within the same model architectures. Our analysis seeks to provide a comprehensive overview rather than a conclusive judgment between these different approaches.

Training LLMs and boosting their performance are pivotal in AI research. However, delving into the fundamental laws and characteristics of LLMs holds equal importance. This comprehensive approach not only provide insight to develop more effective models but also enriches our understanding of the principles underlying their functionality. Our research contributes in two significant ways: First, in LLM training, it dispels the myth that bigger always means better, showing that smaller models can perform on par with, if not outdo, their larger counterparts. This finding prompts a reevaluation of the necessity for continually increasing model size. Our finding also prompts a reevaluation of the rush towards ever-larger models, advocating for a judicious use of REHF with a note of caution on its broad applicability. Second, our analysis sheds light on the critical role of grasping these inherent features in advancing beyond mere performance optimization, towards a more nuanced appreciation of LLMs' capabilities and limitations. For instance, the laws of LLMs tend to be task-dependent. Additionally, we have a deeper understanding of the relationship between different abilities from LLMs, which was not previously explored.

## E. Results of the supplementary dataset

The supplementary dataset, which includes evaluation results of various LLMs, is accessible at `https://docs.google.com/spreadsheets/d/1kT4or6b0Fedd-W_jMwYpb63e1ZR3aePczz3zlbJW-Y4/edit?usp=sharing`. This dataset complements our primary dataset by providing additional information on training parameters and architectures. It encompasses models such as Flan, Pythia, GeoV, codegen, and ChatGPT, which were not included in our primary analysis. Furthermore, it features scores from five evaluation datasets: Lambada, Hellaswag, Winogrande, Piqa, and Coqa.

Each evaluation dataset is unique in its composition and testing methodology. For instance, 'Lambada' contains 5153 passages sourced from books, where the model's task is to predict the final word of each passage based on its preceding context. 'Piqa', on the other hand, comprises 1838 multiple-choice questions that test a model's commonsense physical intuition. The performance of the models on these datasets was measured using accuracy scores for Lambada, Hellaswag, Winogrande, Piqa, and F1 scores for Coqa. Further details can be found in the accompanying article `https://medium.com/@waiyan.nn18/understanding-and-benchmarking-evaluation-me\trics-of-large-language-models-llms-in-2023-9a4a858f782b`.

Our statistical analysis involved applying ANOVA and Tukey's tests to examine the significance of data concerning parameter range scales and model architectures. This approach is consistent with how we processed the primary dataset. The architecture models in this study include 15 frameworks such as Bloom, Falcon, GPT-2, GPT-J, GPT-Neo, Llama, Mistral, OPT, Rwkv, ChatGPT, Flan, Pythia, GeoV, and codegen. The parameter ranges were segmented into five equal ranges: [0.7, 2.8], [2.8, 6.9], [6.9, 11], [11, 15.58], and [15.58, 135].

Significant findings were identified using a significance threshold defined by a p-value less than 0.05. Our analysis across various datasets indicated statistical significance varies across parameter ranges. Notably, the ranges [11, 15.58] and [15.58, 135] showed slight significance in 'Coqa_f1' scores. However, no significant results were found in other evaluation datasets through ANOVA tests. Additionally, models such as 'Llama', 'Pythia', 'GPT-2', 'GPT-J', 'GPT-Neo', and 'OPT' showed significant results in evaluating the datasets.

We applied a similar GAMM fitting to explore the effect of parameters on evaluation scores for each dataset: $(log\_Lambada \sim s(log\_Param) + s(Architecture, bs = "re"), data = data)$. The results, visualized in Fig. 12, indicate that the effects of parameter range scales appear to be linear, without a flattened line at the initial stage, and without a predictable wave-shaped curve when the parameter size increases significantly. However, the curve trend in each plot is identical in Fig. 12, showing that the abilities keep increasing as the parameter becoming larger, and it is the same as "emergent abilities" described in Schaeffer et al. (2023). However, these observations on continuously increasing abilities are merely based on the 65 LLMs in the supplementary dataset. We hypothesize that with a larger number of LLMs, the curve trend could replicate those observed in Fig. 3.

We also employed GAMMs to investigate the interplay between different abilities of language models. The GAMM framework was utilized to understand how one specific ability impacts others. The GAMM equation we applied is: $(log\_Lambada \sim s(log\_HellaSwag) + s(log\_Winogrande) + s(log\_Piqa) + s(log\_Coqa) + s(Architecture, bs = "re"), data = data)$. For each evaluated ability, we formulated a similar GAMM equation with random effects for estimation. In cases where we focused on a specific ability as the dependent variable, that particular ability was omitted from the independent variables in the model. The outcomes of this analysis are depicted in Fig. 13. Our findings reveal that the abilities tested in HellaSwag and Winogrande exhibit a notable influence on the performance in other abilities. This pattern of influence was not observed with the other abilities under consideration. This trend mirrors the observations from our primary dataset, where HellaSwag also demonstrated a general impact on the other abilities of the language models. Moreover, "Lambada" represents text generation has an overall impact on other abilities, which is consistent with the finding of language modeling (text generation) in Burnell et al. (2023).

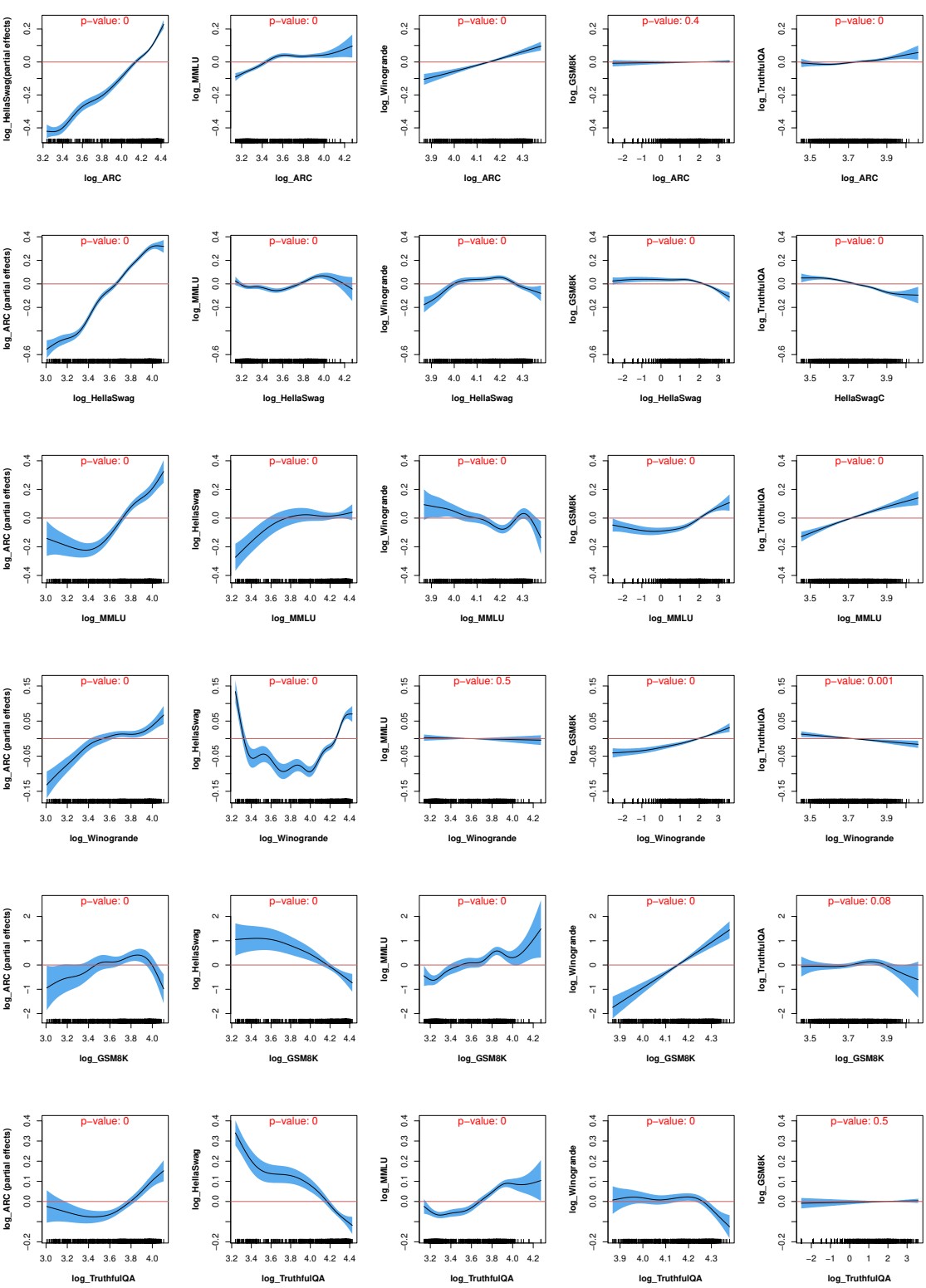

Figure 6: Partial effects of one given ability on other abilities in LLMs. The *x-axis* represents the logarithmic scale of a specific ability, while the *y-axis* corresponds to the logarithmic scale of various other abilities. The slope of the curve provides insights into the strength of this relationship: a steeper slope indicates a more pronounced effect, whereas a gentler slope suggests a more subdued impact. Notably, when the value of $p$ is less than 0.01, the curve tends to level off near zero. This phenomenon signifies that the ability in question has little to no influence on the other ability.

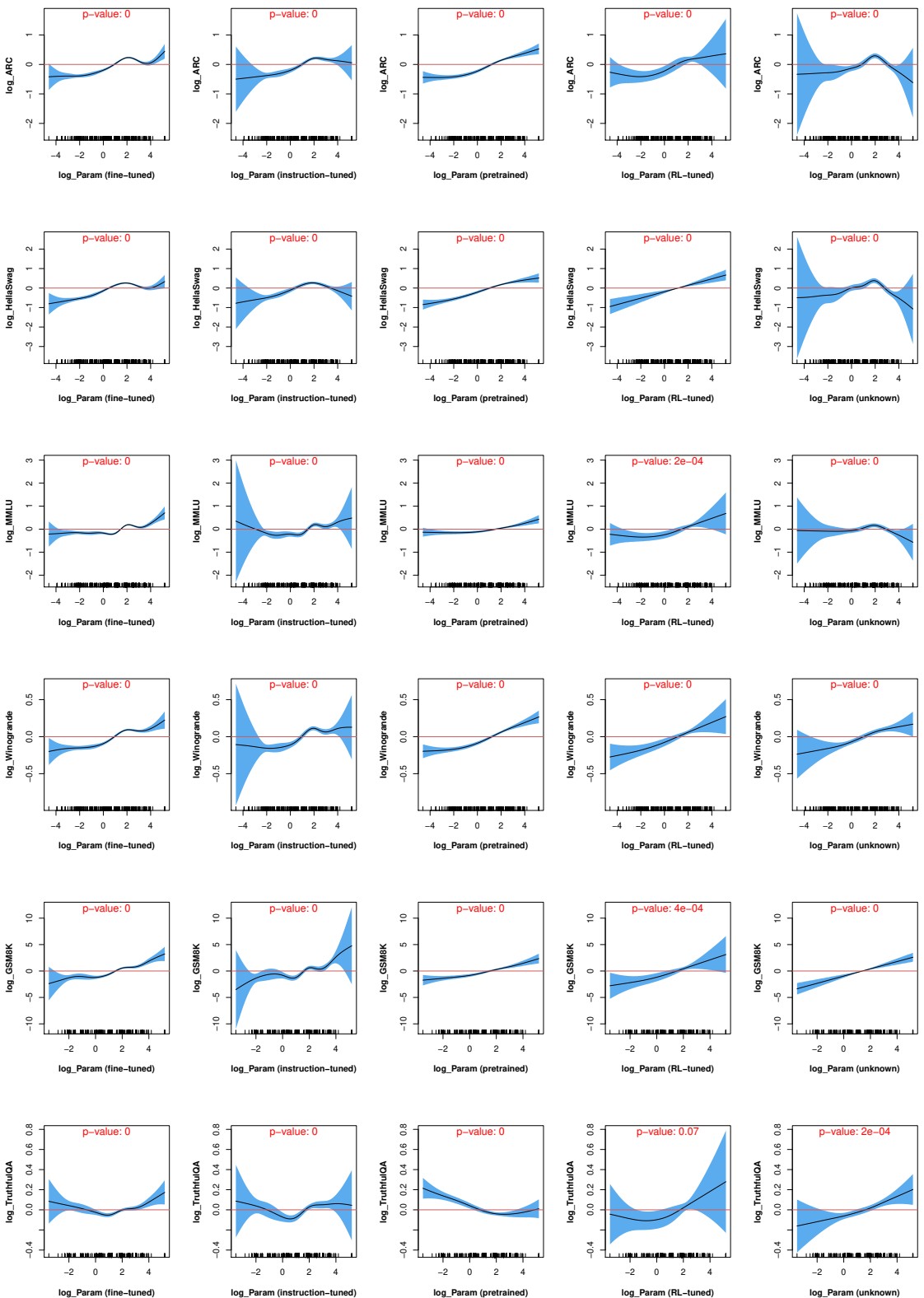

Figure 7: Partial effects of parameters on LLMs performance scores across various training types

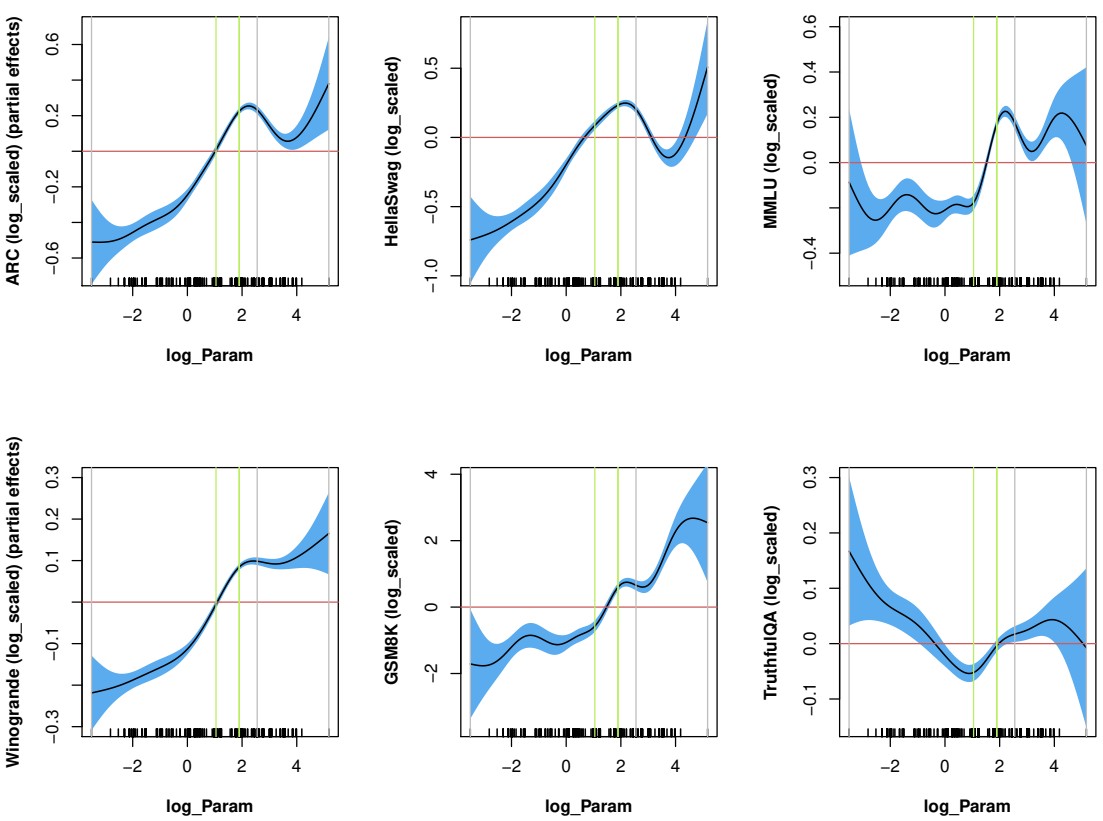

Figure 8: Partial effects of parameters on LLMs performance scores on scaled data

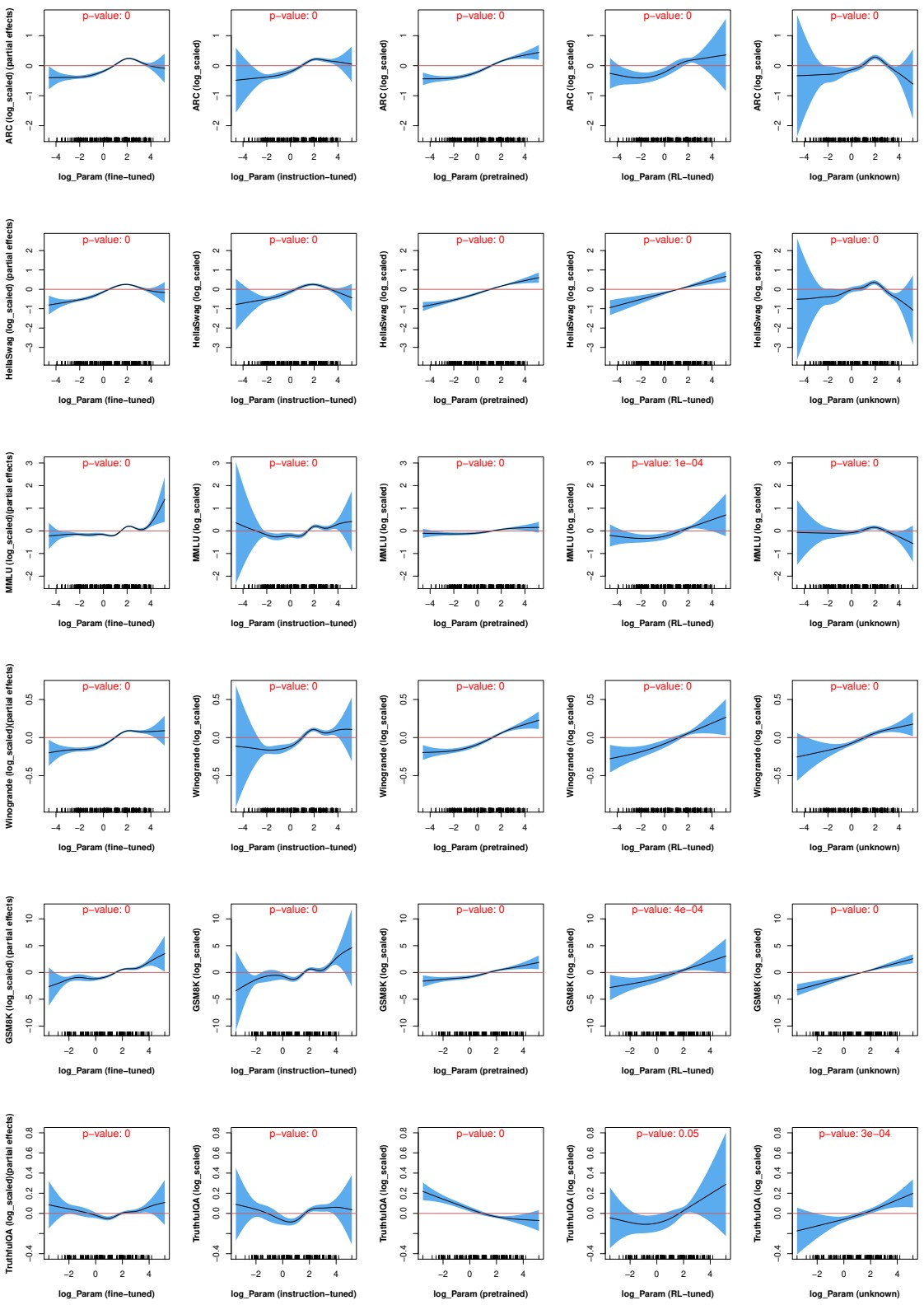

Figure 9: Partial effects of one given ability on other abilities in LLMs on scaled data

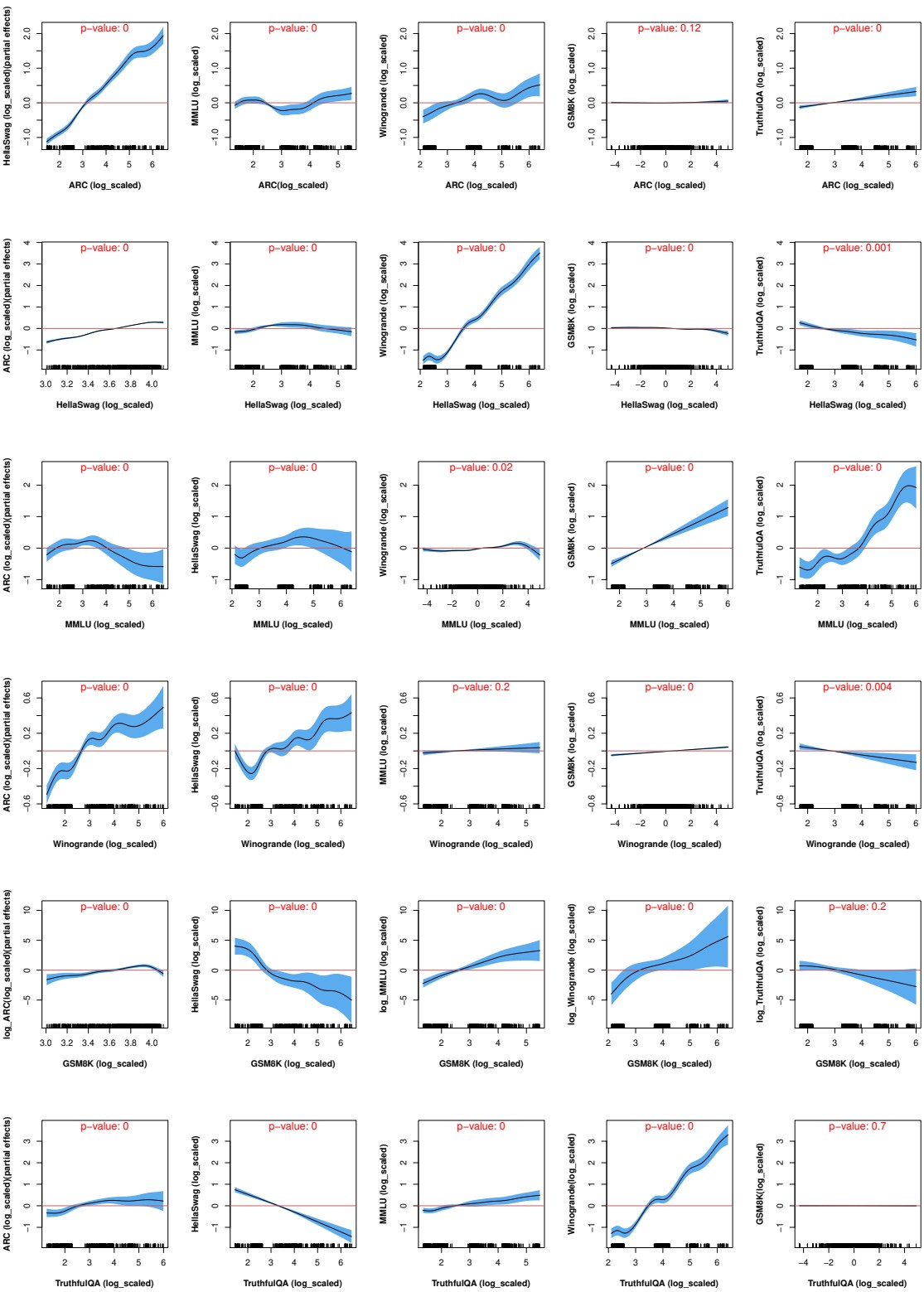

Figure 10: Partial effects of parameters on LLMs performance scores across various training types using scaled data

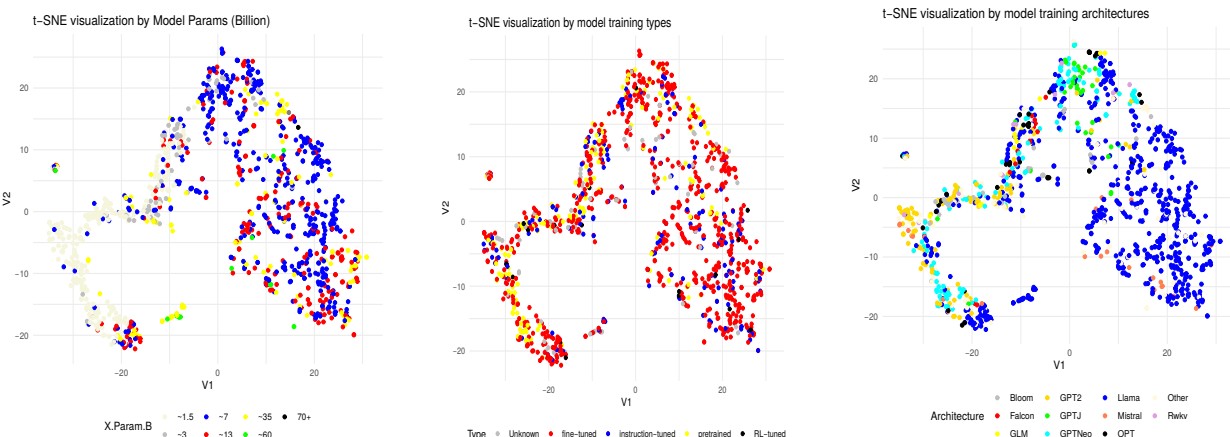

Figure 11: T-sne clusters according to parameter range scale, training types, and architectures

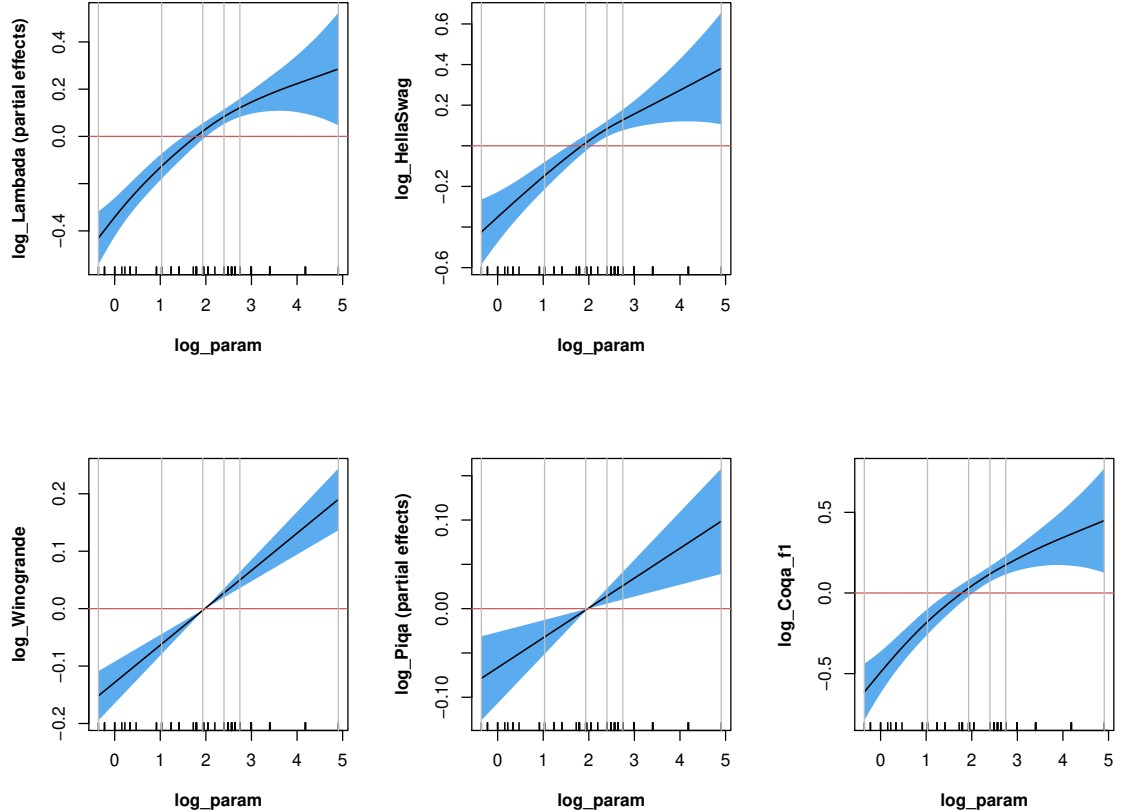

Figure 12: Partial effects of parameters on LLMs performance scores from the supplementary dataset

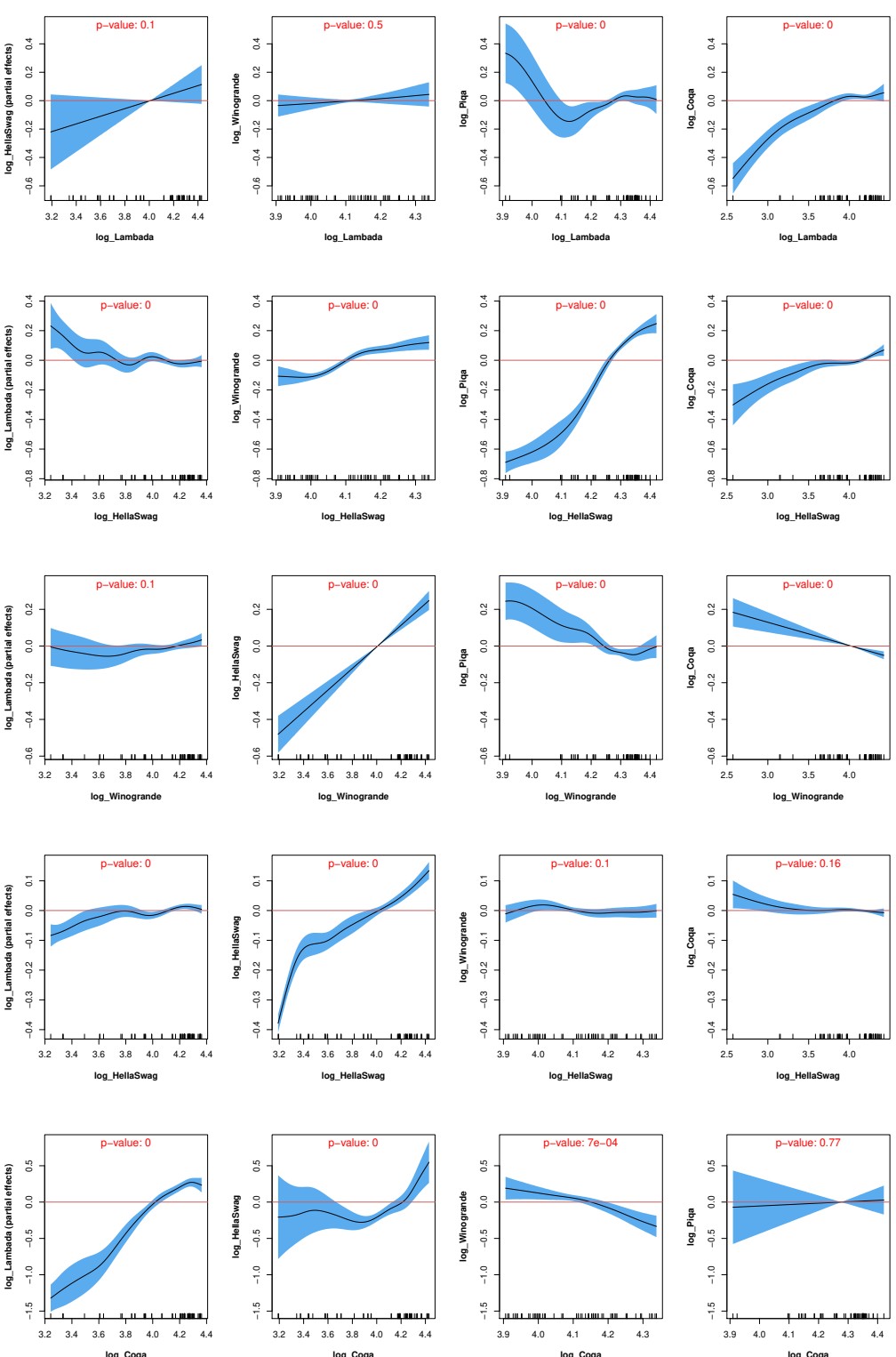

Figure 13: Partial effects of one ability of LLMs on the other abilities in the supplementary dataset

