# OpenReview forum: "Comprehensive Reassessment of Large-Scale Evaluation Outcomes in LLMs: A Multifaceted Statistical Approach"
_TMLR — Rejected by TMLR_

### Review · Reviewer_6rvG · 2024-10-24

**Summary Of Contributions:**

There are a few
- A large scale analysis of over a thousand models that exist on Open LLM Leaderboard
- Statistical analaysis of what design choices empirically associate with performance outcomes.
- They make some pretty interesting object level findings about performance and design choices. I wouldn't have predicted them all.

Overall review: Overall, I can easily see how this paper can be valuable, interesting, and unique. I think that some improvements can be made to the paper, but they are mostly about presentation. I would especially strongly encourage the authors to share a codebase to do the type of analysis they do. I would recommend acceptance conditional on some simple improvements.

**Audience:**

Yes

**Claims And Evidence:**

Yes

**Requested Changes:**

R1: I recommend citing and discussing https://arxiv.org/abs/2405.10938

R2: I think this paper could benefit from having a table or something that briefly summarizes key findings.

**Strengths And Weaknesses:**

S1: One thing I like about work like this is how much it helps us just monitor the field. This kind of awareness in general is useful all around for researchers and governance people alike.

S2: I think that finding some non-monotonic trends is particularly interesting and unexpected.

S3: This paper is very unique I think.

S4: I think this paper will be useful and citeable.

W1: I think this paper is significantly less likely to be useful for a lack of code to help people reproduce and extend the analysis.

W2: I think that this paper is on the wordy side on its presentation -- long paragraphs and few short summary bits. I also think that the title is super correct and descriptive but could probably be changed to better showcase the interestingness of the findings.

---

> ### Author Response · Authors · 2024-11-11
>
> We appreciate your positive feedback and the identification of both the strengths and weaknesses of our manuscript. Here are our responses to the comments and requested changes.
>
>
> ### Presentation and Codebase
>
> 1. **Codebase**
>    - We understand the importance of sharing a codebase to facilitate reproducibility and extension of the analysis. We will include a link to the code in the paper ( The code is available at: https://github.com/fivehills/Reevaluating-LLM-performance.git).
>    - This will enable other researchers to replicate our findings and build upon our work.
>
> 2. **Presentation**
>    - We acknowledge the comment about the paper being wordy and will revise it to include shorter paragraphs and more summary bits.
>    - We will also consider revising the title to better showcase the interestingness of the findings while maintaining its correctness and descriptiveness. We will consider a revision that better captures the interesting and unique aspects of our findings.
>     - We will thoroughly revise the paper to improve its presentation, focusing on clarity and conciseness. This includes breaking down long paragraphs into shorter ones and adding summary sections to highlight key points.
>
>
> ### Summarizing Key Findings  and Additional Citation/Discussion
>
> 1. **Summary Table**
>    - We will add a table summarizing the key findings to help readers quickly grasp the main results at a glance.
>  2. **Citing and Discussing Additional Work**
>     - We will cite and discuss the paper recommended (https://arxiv.org/abs/2405.10938) to provide a more comprehensive context and comparison with related work.
>
>
> ### Conclusion:
>
> We greatly appreciate your detailed review and recommendations. Following your suggestions, we plan to enhance the paper by making the codebase available, improving the presentation, summarizing key findings, and adding further discussions. We believe these changes will significantly strengthen the paper’s quality and value for the research community.

---

> > ### Comment · Reviewer_6rvG · 2025-01-03
> > **I recommend acceptance**
> >
> > Thanks. I appreciate the response. I think that it's helpful and the promised chances are good and doable. I think this paper will be citeable and valuable to the ML/TMLR communities.

---

### Review · Reviewer_REqA · 2024-10-24

**Summary Of Contributions:**

The authors analyze several LLMs with respect to three features taken from the Open LLM Leaderboard - architecture, training type, and parameters - looking at performance across several datasets. The authors conduct the analysis with sound statistical methods. The authors' findings notably suggest that increasing parameter count does not necessarily mean performance will increase, challenging popular notions about LLMs.

**Audience:**

Yes

**Broader Impact Concerns:**

None; the paper takes an abstract perspective that does not focus on specific languages, groups, use cases, etc.

**Claims And Evidence:**

Yes

**Requested Changes:**

Critical to securing recommendation:
* Explain what the authors mean by “emergent abilities” - this term has been thrown around a lot in the NLP community without a clear definition (copy the definition from 3.2 in the intro)
* Figure 1 could be improved greatly. Parts of the image are blurry when zooming in. Some text is stretched. Font sizes and appearance vary dramatically. Image does not convey enough information to get a grasp of the message.
* It looks like the contribution of the paper is to analyze performance of LLMs, particularly in terms of evidence of emergent abilities, with respect to a variety of previously under-considered factors, including differences in training and architecture. If this is correct, this message needs to be made much clearer in the introduction. In particular, because this work sits in the midst of a variety of related work, it would be valuable to highlight the specific new factors being considered and/or the contributions of this work in a concise form, preferably outside of the main prose (e.g., via bullet points or in its own paragraph with some bold).
* Is this taxonomy of architectures, training types, etc. through which the authors analyze LLMs novel work or drawn from past work?
* What is the “data” in this study? (referenced multiple times in Section 2)
* How are the dependent, independent, and random variables selected? What role does each of these categories play in the analysis (especially random variables)? Clarifying this would strengthen the explanations in this section. Perhaps specific research question(s) could be listed building on these variables.
* Importantly, how is this work separated from Open LLM Leaderboard? Certain things are described in the paper in a way that makes the reader think it is a novelty of the paper but rather they may be from Open LLM Leaderboard. This distinction could be made more clearly throughout the paper.
* Lots of strong analysis; however, discussion of findings and implications is a little murky. It would be helpful to make the discussion much stronger as well, particularly in terms of understanding why certain differences might be present along the different axes of variation, and the implications for LLM selection, usage, and training.

Simply strengthen the work:
* ”It is prudent to establish..." at the beginning of Section 2 is a little misleading, since it has already been created in the previous work listed.
* Should include BIG-Bench (Srivastava et al., 2023) from TMLR with the list of citations in the first paragraph of the paper.
* "scaling law Kaplan et al. (2020)" --> scaling law (Kaplan et al., 2020)
* The metaphor with human cognitive abilities is interesting but flawed (at least without much more explanation). It may be better to leave it out.
* Several spaces are missing, especially preceding in-text citations. Additional typos as well (e,g., “aht” —> that).
* What is meant by “with their proportions”? The percentages in 2.1 are unclear.
* “we cross-validated the other data on LLMs evaluation results…” sentence does not make sense. Also, if that is the supplementary dataset, what is the main dataset?
* “The following details the three statistical…” sentence does not make sense.
* In 2.2, giving examples each time (e.g., ARC, HellaSwag) makes it difficult to read and understand.
* In 3.1, the list of 12 frameworks already has 10. Instead of “and others,” finish the list.
* Figure 2 is blurry and text is too small.
* 3.2 contains a lot of details but some are not that clearly presented. It would be beneficial to rework the section to make it more readable.
* “than abrupt breakthroughs (?)” - missing citation
* “This study provides new perspectives…contributing a unique perspective…” - redundant

**Strengths And Weaknesses:**

Strengths:
The paper includes lots of technical details and multiple methods of statistical analysis, and shows these measures across several models and datasets. The experiments of the paper are self-contained and may yield a good output. The weaknesses focus on the written composition, particularly, with how the results are utilized to understand the problem and paint a bigger picture, without which the results are not as meaningful.

Weaknesses (written in more detail in the request changes section):
* The clarity in writing is not always at its peak. Increasing clarity, especially in terms of contributions (introduction) and key findings/implications (discussion), would be extremely beneficial to the quality of the paper.
* Beyond my understanding conveyed in summary of contributions (primarily, analysis), it is unclear how this work separates from past work like Open LLM Leaderboard.
* Certain choices for the structure of analyses seem arbitrary (particularly, choice of independent, dependent, and random variables). The features for each of these are taken from Open LLM Leaderboard but not much motivation is provided for why they are the best features to consider.
* Many findings could be synthesized to paint a much clearer and bigger picture of what's going on. As of now, the several tests across models and datasets are kind of their own isolated findings, and the big picture seems to just be that increasing parameter count does not necessarily mean performance will increase. Whether this is the only key finding or there are more key findings, with so much analysis, I am sure they could be synthesized to inform the overall findings more to increase relevance for readers.

---

> ### Author Response · Authors · 2024-11-11
>
> Many thanks for your detailed feedback, which we greatly appreciate. We acknowledge both the strengths and weaknesses identified in the review. We will address your feedback by making the necessary corrections and clarifications.
>
> ### Clarity and Contributions
> To address the clarity issues, we will revise the introduction and discussion sections to better articulate the contributions and key findings of our study.
>
> 1. **Clearer Introduction**
>    - We will explicitly define what we mean by “emergent abilities” in the introduction, drawing from the relevant section (3.2) to provide clarity.
>    - We will highlight the specific new factors being considered in our analysis, such as differences in training types and architectures, and how these contributions distinguish our work from previous studies. This will be presented in a concise form.
>
> 2. **Key Findings and Implications**
>    - We will synthesize the findings to paint a clearer and bigger picture of the overall results. This includes explaining why certain differences might be present along the different axes of variation and the implications for LLM selection, usage, and training.
>    - We will ensure that the discussion of findings and implications is stronger, providing a more cohesive understanding of the results and their significance.
>
> ### Structure and Motivation of Analyses
>
> 1. **Selection of Variables**
>    - We will provide a clear motivation for why we chose the features of architecture, training type, and parameters from the **Open LLM Leaderboard**. This includes explaining the role of each category (independent, dependent, and random variables) in our analysis.
>    - We will list specific research questions that our analysis aims to address, building on these variables.
>
> 2. **Distinction from the Open LLM Leaderboard**
>    - This may be your biggest concern on our contributions. Here we clarify how our work differs from the _Open LLM Leaderboard_. The **Open LLM Leaderboard** serves as a platform that aggregates LLM performance information across several test datasets using consistent evaluation standards, with data provided voluntarily by developers. Although the **Leaderboard** allows users to view model performances, it does not directly provide insights on model differences across architectures or training types, nor does it reveal patterns like non-linear relationships or emergent abilities as parameters scale. By applying multi-faceted statistical analysis to this raw data, our approach uncovers these deeper insights and findings, which cannot be derived from the **leaderboard** alone.
>    - We will add some information to distinguish what we can get from the **Open LLM Leaderboard**, ensuring that the reader understands the unique contributions of our research.
>
> ### Data and Statistical Terms
>
> 1. **Data Description**
>    - We will clarify what is meant by “data” in Section 2, ensuring that it is clear whether we are referring to the main dataset from the Open LLM Leaderboard or the supplementary dataset.
>    - We will correct the sentence regarding cross-validation to make it clear that the supplementary dataset is used for validation purposes.
>
> 2. **Statistical Methods**
>    - We will ensure that the explanation of our statistical methods is clear and concise. For example, we will avoid repetitive examples (e.g., ARC, HellaSwag) and focus on providing a comprehensive overview of the methods used.
>
> ### Figures and Formatting
> To improve the clarity and readability of the figures:
>
> 1. **Figure 1**
>    - We will revise Figure 1 to ensure it is not blurry when zoomed in, correct the font sizes and appearance, and make sure the image conveys the intended message clearly.
>
> 2. **Figure 2**
>    - We will also improve Figure 2 by ensuring it is not blurry and that the text is readable.
>
> ### Miscellaneous Corrections
> To address the formatting and citation issues:
>
> 1. **Typos and Spaces**
>    - We will conduct a thorough review to correct typos, missing spaces, and ensure proper formatting, especially preceding in-text citations.
>
> 2. **Citations and References**
>    - We will include BIG-Bench (Srivastava et al., 2023) in the list of citations in the first paragraph.
>    - We will correct the citation format for “scaling law Kaplan et al. (2020)” to “scaling law (Kaplan et al., 2020)”.
>    - We will remove or clarify the metaphor with human cognitive abilities if it is deemed flawed.
>
> 3. **Proportions and Percentages**
>    - We will clarify the percentages and proportions mentioned in Section 2.1 to ensure they are clear and understandable.
>
> 4. **Section Readability**
>    - We will rework Section 3.2 to make it more readable and ensure that all details are clearly presented.
>    - We will complete the list of frameworks in Section 2.1 instead of using “and others.”
>
>
> We really appreciate your constructive suggestions. We believe that addressing these concerns will significantly enhance the quality and impact of our manuscript.

---

> > ### Comment · Reviewer_REqA · 2024-11-21
> > **Details Regarding Some Response Points**
> >
> > Thanks for the well-structured response. Before the discussion period ends, if the authors have a chance, it would greatly aid my ability to evaluate the paper if details can be provided for a few of the points made in the authors' response. Specifically, what is that clear, irrefutable motivation you plan to provide regarding the variables (Structure and Motivation of Analyses > Selection of Variables), and what will the rewrite of Key Findings and Implications (Clarity and Contributions) look like, briefly? Thanks.

---

### Review · Reviewer_fXYp · 2024-11-09

**Summary Of Contributions:**

This paper assembles a dataset of LLM benchmark results for a whole lot of LLMs and a whole lot of benchmarks, and then does analyses to determine the extent to which various LLM properties affect the results of the benchmarks.

**Audience:**

No

**Claims And Evidence:**

No

**Requested Changes:**

- Improve clarity throughout (critical)
- Justify why this kind of methodology makes sense, as opposed to doing more direct comparisons.

**Strengths And Weaknesses:**

Overall, I am not convinced that the core methodology of this paper is promising. The basic idea of this paper is to "zoom out" and compare broad classes of LLMs to each other, with the goal of learning about general patterns of differences between LMs. For example, the paper compares fine-tuned to instruction-tuned LMs. I think that it's probably a mistake to do this by comparing a huge number of LMs, instead of the more standard methodology of doing close comparisons (e.g. between training the same base model with both fine-tuning and instruction-tuning). The methods in this paper are often used in fields like social science where it's not possible to do intervention experiments; I think that they aren't applicable here.

Doing this kind of experiment means that you're exposed to all sorts of sources of error and noise. For example, there are lots of poorly trained LMs on Huggingface; you're exposed to the bias where the poorly trained models systematically have one type of architecture.

So overall I didn't learn much from this approach.


I also found this paper quite hard to understand. There were many grammatical errors and sentences that didn't make sense. I also found the structure of the paper sort of confusing.

---

> ### Author Response · Authors · 2024-11-11
>
> Thanks for your feedback and the suggestion to use more direct method. While we see the value in this approach, it is not directly aligned with our core methodology and findings. We address your concerns on methods and data as follows.
>
>
>
>
> ### Comparison of Broad Classes of LLMs
>
> 1. **Generalizability **: Our large-scale analysis of LLMs identifies patterns that aren’t evident in smaller comparisons. This broad statistical approach allows generalizations across a diverse set of models, essential given their rapid evolution. Moreover, developers often choose from a wide range of base models. Our approach mirrors this real-world context, providing insights relevant for LLM selection and deployment.
>
> 2. **Traditional Approach**: While direct comparisons (e.g., fine-tuning vs. instruction-tuning based on the same model) are standard, real-world scalability of such paired data collection is limited. If instruction-tuning is genuinely beneficial, its effects should be observable across diverse base models.  Our initial descriptive statistical analysis, without direct comparisons you mentioned, shows significant differences between both instruction-tuning and fine-tuning vs. base model, but no significant difference between instruction-tuning and fine-tuning. If our method were ineffective, we wouldn’t observe these distinctions.  The insignificance  case is not due to our approach. Actually our statistical analysis across diverse LLMs tests this with rigor, capturing wider trends essential to assessing techniques’ robustness.
>
> 4. **Core Statistical Analysis**: We used ANOVA for initial comparisons between fine-tuning and instruction-tuning, finding no significant differences, which raises questions about instruction-tuning’s effectiveness. However, this comparison was not the study’s primary focus. Our core analysis, using Generalized Additive Mixed Models, investigated non-scaling laws, gradual capability emergence, and ability interactions in LLMs. Such comparisons (e.g., fine-tuning vs. instruction-tuning) are **not related with our core methods and main findings**. For example, in Fig. 3, we analyze the relation between specific abilities and training parameters without fine-tuning vs. instruction-tuning distinctions (see additional findings in Supplementary Figs. 7, 10, 12).
>
> In sum,  you raised questions on methodology, but have not actually challenged our core statistical approach or main findings.
>
> ### Wider Applicability of Statistical Methods
>
> - Statistical methods are not confined to social sciences but are widely applied in NLP,  ML, and DL, as shown in recent monographs like “Validity, Reliability, and Significance” (Riezler, et al. 2024) and “Statistical Significance Testing for NLP” (Dror et al., 2020). These methods extend across scientific fields—biology, medicine, cognitive science—where they are critical for analyzing complex, multi-variable systems.
>
> - We can view the LLM population as an ecosystem or population, much like biological systems where interactions and dynamics are studied through statistical analysis. This approach reveals patterns across model architectures and training methods, much like ecologists uncover species interactions and emergent behaviors.
>
> - Moreover, analyzing LLMs shares parallels with cognitive science, where statistical rigor is essential for understanding cognitive abilities and patterns. As LLMs move toward AGI, evaluations should mirror the approaches used to study human cognition. Simple comparisons offer quick insights but risk overlooking real patterns that statistical methods could reveal. Evaluating LLMs from a broader, population-level perspective—beyond specific technical details—can discover realistic trends. Our approach connects high-level patterns with specific technical factors, deepening our understanding of what drives LLM performance and guiding more credible advancements toward AGI.
>
> ### Data Quality
>
> 1. **Real-World Representation**: The **Open LLM Leaderboard** data offers a realistic snapshot of LLM performance in practical applications, including the variability and errors present when deployed in diverse environments. Unlike research focusing on a few controlled models to limit noise, this broader dataset captures the rapid growth, diversity, and real-world challenges of LLMs more effectively.
>
> 2. **Data**: Not all Hugging Face models appear on the *Open LLM Leaderboard*. The dataset includes about 1,100 models based on 12 major popular architectures. Since numerous submissions to the **Leaderboard** are voluntary, researchers are more likely to share significant results. In contrast, research relying on direct comparison introduces more bias, as it selectively chooses a few given models, test datasets, and evaluation standards, often reporting only favorable results from a single submission. Compared to the **Leaderboard**, results from such methods are difficult to reproduce.

---

### Decision · Action_Editor_16G6 · 2025-01-09

**Recommendation:** Reject

**Comment:**

I apologize for the delay in getting the decision out for this paper. The authors have clearly made a genuine effort, but the reviewers' recommendations were split (one accept, one borderline accept, and one borderline reject), and based on the content of the paper, the reviews, and the subsequent reviewer-author discussion, I feel it is difficult to justify acceptance of the current paper for TMLR. The [key acceptance criterion](https://jmlr.org/tmlr/acceptance-criteria.html) for TMLR is that the paper's main claims are clear and supported by solid evidence, and as I mentioned earlier, I think this paper lacks a well-supported main claim, and reads almost like an internal-use technical report.

If the authors want to put forward a new method for evaluating LLMs, that's one perfectly natural option, but it must be described with sufficient generality and compared with relevant approaches in the literature. If on the other hand the authors want to emphasize some new empirical findings, the experimental conditions need careful consideration. Since at present it is the choice of experimental conditions that seems to characterize this work, while the findings may strictly speaking be "new," their relation to and impact on prevailing empirical insights from the literature (based on different, more sharply controlled conditions) is in my opinion limited.

**Audience:**

Analysis of LLMs is something every other person seems to be doing these days, and while new viewpoints and methodologies are assuredly sought by the LLM and machine learning community at large, the presentation issues compounded with a lack of a substantive point which can be considered a "main claim" means that to me, the effective audience for this paper is quite limited.

**Claims And Evidence:**

I personally found the main claim(s) of this paper rather difficult to parse, even after reading through the reviews. I understand that the authors have aggregated performance data for a wide variety of LLMs via the Open LLM Leaderboard (among other sources?), and passed this data through a handful of statistical procedures (e.g., ANOVA, GAMM approximation, t-SNE). I also undertstand that based on this analysis, the authors are emphasizing that the simple story of "more parameters leads to better performance" is not necessarily true, but as highlighted by one reviewer, by piling in a sufficiently wide variety of LLMs (including those which have been poorly trained), all sorts of convoluted statistical trends could arise to confuse the data scientist.

If the main claim is supposed to be for the proposal of a new general-purpose method for evaluating LLMs, then as the reviewers also note, the quality of writing and overall presentation is too low to justify the paper's content as "convincing evidence" for such a method. Section 2 should be where this method is described clearly, but it just branches into descriptions of data selection from the "Leaderboard" and existing statistical procedures the authors have elected to use; a new underlying "method" is not clear to me.

**Resubmission Of Major Revision:**

The authors may consider submitting a major revision at a later time.